**www.cambridge.org/ext**

## Overview Review

defaunation; prehistoric *Homo sapiens*; megafauna extinctions; Pleistocene climate; trophic rewilding

**Corresponding author:**
Jens-Christian Svenning;
Email: svenning@bio.au.dk

# The late-Quaternary megafauna extinctions: Patterns, causes, ecological consequences and implications for ecosystem management in the Anthropocene

Jens-Christian Svenning ⓘ, Rhys T. Lemoine ⓘ, Juraj Bergman ⓘ, Robert Buitenwerf ⓘ, Elizabeth Le Roux ⓘ, Erick Lundgren ⓘ, Ninad Mungi ⓘ and Rasmus Ø. Pedersen ⓘ

Center for Ecological Dynamics in a Novel Biosphere (ECONOVO) & Center for Biodiversity Dynamics in a Changing World (BIOCHANGE), Department of Biology, Aarhus University, Aarhus, Denmark

## Abstract

Across the last ~50,000 years (the late Quaternary) terrestrial vertebrate faunas have experienced severe losses of large species (megafauna), with most extinctions occurring in the Late Pleistocene and Early to Middle Holocene. Debate on the causes has been ongoing for over 200 years, intensifying from the 1960s onward. Here, we outline criteria that any causal hypothesis needs to account for. Importantly, this extinction event is unique relative to other Cenozoic (the last 66 million years) extinctions in its strong size bias. For example, only 11 out of 57 species of megaherbivores (body mass ≥1,000 kg) survived to the present. In addition to mammalian megafauna, certain other groups also experienced substantial extinctions, mainly large non-mammalian vertebrates and smaller but megafauna-associated taxa. Further, extinction severity and dates varied among continents, but severely affected all biomes, from the Arctic to the tropics. We synthesise the evidence for and against climatic or modern human (*Homo sapiens*) causation, the only existing tenable hypotheses. Our review shows that there is little support for any major influence of climate, neither in global extinction patterns nor in fine-scale spatiotemporal and mechanistic evidence. Conversely, there is strong and increasing support for human pressures as the key driver of these extinctions, with emerging evidence for an initial onset linked to pre-*sapiens* hominins prior to the Late Pleistocene. Subsequently, we synthesize the evidence for ecosystem consequences of megafauna extinctions and discuss the implications for conservation and restoration. A broad range of evidence indicates that the megafauna extinctions have elicited profound changes to ecosystem structure and functioning. The late-Quaternary megafauna extinctions thereby represent an early, large-scale human-driven environmental transformation, constituting a progenitor of the Anthropocene, where humans are now a major player in planetary functioning. Finally, we conclude that megafauna restoration via trophic rewilding can be expected to have positive effects on biodiversity across varied Anthropocene settings.

## Impact statement

Terrestrial large-bodied animals (megafauna) play important roles in ecosystems and human cultures. However, their diversity and abundance have declined severely across the last ~50,000 years. This late-Quaternary megafauna extinction pattern stands out from previous Cenozoic extinctions in three ways. (1) These losses were global and severe. (2) They were strongly biased toward larger-bodied species, with other organisms experiencing only very limited extinction in this period. Illustrating this pattern, only 11 out of 57 species of megaherbivores (mean body mass ≥1,000 kg) survived through to 1,000 AD. (3) This faunal simplification is unique on a ≥30-million-year time scale, with diverse megafauna guilds being the norm throughout this entire timeframe, excepting recent millennia. Further, temporal staggering is a defining feature of these losses, with extinctions concentrated in widely different time windows in different areas. The debate on the cause, or causes, of the late-Quaternary extinctions has been ongoing for over 200 years. Though most current work accepts at least a contributory role for modern humans, the topic remains controversial. We outline multiple characteristics of the late-Quaternary extinctions that, in order to merit support, any hypothesis needs to account for, and based thereon conclude the existing evidence strongly supports a dominant role of *Homo sapiens* and is inconsistent with climate as a substantial cause. We discuss the known and likely ecological consequences of the late-Quaternary megafauna extinctions, with the combined evidence indicating that the disappearance of so many large animal species constitutes a fundamental re-shaping of terrestrial ecosystem worldwide.

Ecosystem effects can be grouped into trophic processes, physical environmental engineering and the transportation of energy and matter. Building thereon, we outline megafauna-based trophic rewilding as a key approach to restoration under global change. We also discuss the interplay of megafauna and human-driven biotic globalization and the ecological problems and potential in domestic megafauna, that is, livestock.

## Introduction

There is a deep and ancient connection between humans and large animals (megafauna), which persists to this day (Berti and Svenning, 2020). This relationship has at times been adversarial, at others reverential, and at all times, due to the reliance of humans on animal-based resources, practical (e.g., Ben-Dor et al., 2011; Ripple et al., 2016; Domínguez-Rodrigo et al., 2021; Gaudzinski-Windheuser et al., 2023; Smith et al., 2024). Respect for and value of the largest fauna by modern humans (*Homo sapiens*) is evident in prehistoric art across the world, which often predominantly depicts the largest animals that humans would have encountered (e.g., Malotki and Wallace, 2011; Hussain and Floss, 2015; Zeller and Göttert, 2021). Unfortunately, this importance to people would appear to have been historically detrimental, as the most massive animals consistently disappear following human arrival or intensified occupation (e.g., Martin, 1967; Surovell et al., 2005; Koch and Barnosky, 2006; Teng et al., 2020; Dembitzer et al., 2022). The ongoing loss of megafauna worldwide is not only a conservation issue but also an ecological issue, given the increasing evidence that large animals play important roles for biodiversity and in ecosystem functioning (e.g., Estes et al., 2011; Malhi et al., 2016; Galetti et al., 2018; Enquist et al., 2020). As such, there is an increasing focus on large animals in conservation, restoration and climate change mitigation and adaptation (e.g., Svenning et al., 2016, 2024; Cromsigt et al., 2018; Malhi et al., 2022). Despite this strong scientific interest in large-bodied animals, there is continued discussion around not just their role in the biosphere, but also their past and present relationship with humans.

One issue on which there is much discussion is the strong downsizing of terrestrial vertebrate assemblages across the last ~50,000 years, due to severe extinctions and extirpations of the larger species (Martin, 1967; Smith et al., 2018). With the advent of $^{14}$C dating, it has become clear that Earth suffered widespread, severe extinctions among its terrestrial megafauna in recent prehistory, specifically during the Late Pleistocene (129,000–11,700 years BP) and Early to Middle Holocene (11,700–4,200 years BP), hereafter referred to as the late-Quaternary extinctions (e.g., Martin, 1966). This pattern stands out from previous Cenozoic extinctions in three ways. First, the losses were global and severe: prior to this event, mainland faunas consistently harbored diverse assemblages of large mammals, while island systems had rich faunas of medium-large mammals, birds, and reptiles (e.g., Stuart, 2015). On the continents, communities shifted from highly diverse megafauna assemblages that included proboscideans and other megaherbivores (≥1,000 kg) as well as a range of megacarnivores (≥100 kg) to communities with few or no such species (e.g., Stuart, 2015; Malhi et al., 2016). Second, other organisms were not similarly affected, with plants, invertebrates and small vertebrates experiencing only very limited extinction (e.g., Raffi et al., 1985; Stuart, 1991; Magri et al., 2017), the exception being megafauna-dependent organism groups such as scavenging birds and dung beetles (Galetti et al., 2018). Third, this simplification of the faunal community is unique on a 30-million-year or deeper time scale, with diverse megafauna guilds being the norm throughout this entire timeframe save for, depending on the region, the last 50,000–2,000 years (Smith et al., 2018). The cause of this extreme downsizing has been the subject of active study and debate since the 1960s (e.g., Martin, 1966, 1967; Koch and Barnosky, 2006; Stuart, 2015). While there are multiple causal hypotheses, the emphasis has been on the evolution and expansion of modern humans (*Homo sapiens*) (Martin, 1967; Sandom et al., 2014a; Bartlett et al., 2016; Araujo et al., 2017) and on climatic pressures associated with the last glacial–interglacial cycle (Cooper et al., 2015; Carotenuto et al., 2016; Stewart et al., 2021).

Megafauna losses have continued up through the latter part of the Holocene to the present. Earth's remaining megafauna are in quite a dire state, with ~47% of all living mammals weighing ≥10 kg listed as vulnerable, endangered or critically endangered by the International Union for the Conservation of Nature (IUCN), and an additional ~12% listed as near-threatened (calculated with PHYLACINE (Faurby et al., 2018, 2020a)). Defaunation is widespread and expanding in many parts of the Global South (Dirzo et al., 2014), often preceding habitat destruction. This is encapsulated by Empty Forest Syndrome, wherein forests are still standing but with major ecological dysfunction due to the extirpation or near-extirpation of their larger vertebrates (Redford, 1992). Further, losses of megafauna in the most recent 1–5 millennia are widely reported, including severe declines in the ranges of many extant species. For example, China has seen strong range contractions in elephants (*Elephas maximus*), rhinoceroses (*Dicerorhinus sumatrensis*, *Rhinoceros sondaicus*) and tigers (*Panthera tigris*), as well as the global extinction of a water buffalo (*Bubalus mephistopheles*), an equid (*Equus ovodovi*) and a gibbon (*Junzi imperialis*) in recent millennia (Turvey et al., 2018; Teng et al., 2020). Similar declines in megafauna occurred in Europe (Crees and Turvey, 2014; Crees et al., 2016) and the Middle East (Tsahar et al., 2009; Bar-Oz et al., 2011). The relationship between these losses and intensified human impact is widely accepted, yet classic conservation and restoration focus on a baseline set at or after 1500 CE, a time when ecosystems were already highly simplified or degraded (Martin, 1967; Donlan et al., 2006; Monsarrat and Svenning, 2022).

The history of large-animal faunal dynamics is an increasingly practical consideration, both to inform and implement adequate conservation and restoration policies for endangered megafauna and to provide an informed basis for managing recovering and expanding populations of megafauna. An important counterpoint to widely continuing declines of large animals is the strong expansion of megafauna species observed in various, usually high-income regions. This dynamic is pronounced in Europe, reflecting increased societal tolerance, leading to expansions in most extant, wild-living megafauna species across recent decades (Ledger et al., 2022). This includes, for example, a >16,000% increase in Europe's population of beaver (*Castor fiber*), a >300% increase in red deer (*Cervus elaphus*), a >300% increase in wild boar (*Sus scrofa*), a >16,000% in European bison (*Bison bonasus*) and a >1800% increase in gray wolf (*Canis lupus*) since 1960–1965 (Ledger et al., 2022). There have been similar expansions in North

America, Japan and Australia (Kaji et al., 2000; Martin et al., 2020; Read et al., 2021). This recovery dynamic often includes alien megafauna as well (Lundgren et al., 2018), for example wild pigs (*Sus scrofa*) in much of the Americas (Vercauteren et al., 2020; Hegel et al., 2022). While these recoveries are receiving positive attention, they are also subject to controversy and calls for strong population control, as seen in debates around the overabundance of deer (Côté et al., 2004; Martin et al., 2020) and kangaroos (Read et al., 2021), large carnivore comebacks (Martin et al., 2020; von Hohenberg and Hager, 2022), and alien ungulates (Lundgren et al., 2018). Finally, there is rising interest in the potential role of megafauna for ecosystem-level conservation and restoration efforts, notably in the concept of trophic rewilding (Svenning et al., 2016, 2024).

In this review, we appraise the current state of knowledge on the late-Quaternary extinctions, with focus on the patterns, drivers and consequences of megafauna disappearance as well as its relevancy for conservation and restoration. In palaeobiology, the definition of megafauna typically relies on a body weight threshold ≥45 kg, but other definitions and imprecise usages are common (Moleón et al., 2020). Body mass is arguably the most important functional trait within animals, with strong effects on key aspects of their biology including life cycle, energy usage, and ecological impacts (e.g., Smith et al., 2016; Enquist et al., 2020; Fricke et al., 2022). While

the relationships between these factors and body mass are often non-linear, thresholds are mostly arbitrary and relative impacts can vary depending on biogeographic context with, for example, relatively small animals on isolated islands often functioning as megafauna (Hansen and Galetti, 2009). Consequently, we use a broad definition of megafauna that generally includes all animal species with a typical adult body weight ≥10 kg, but with attention to body size effects and allowing for varying definitions between different studies. Furthermore, we focus on terrestrial megafauna, as Quaternary extinctions among marine megafauna are much fewer and clearly attributable to recent human impacts.

## Extinction patterns

The broad-scale spatiotemporal patterns in the late-Quaternary extinctions are well described (Figure 1; e.g., Martin, 1967; Koch and Barnosky, 2006; Smith et al., 2018; Lemoine et al., 2023). Among terrestrial mammals, only 11 out of 57 species of megaherbivores (mean adult body mass ≥1,000 kg) survived through to 1,000 AD, that is, an 81% extinction rate (Table 1). The survivors include three species of elephant, four species of rhinoceros (the critically endangered fifth species, the Sumatran rhinoceros (*Dicerorhinus sumatrensis*), only has a 700–800 kg body mass), the common hippopotamus (*Hippopotamus amphibius*), and, at close to the

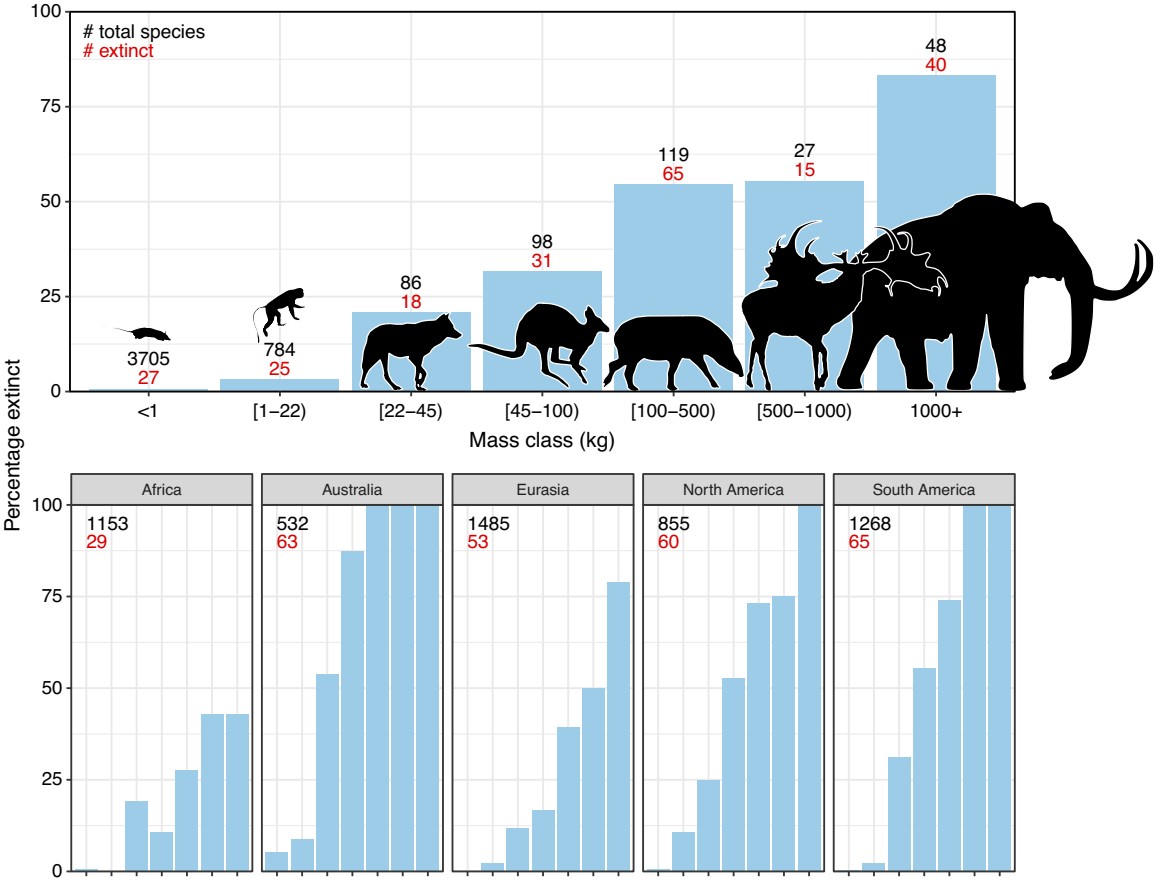

**Figure 1.** Late-Quaternary mammal extinctions as a function of body size. The global proportion of extinct species as a function of body size is shown at the top, and split per continent at the bottom. Black numbers are total late-Quaternary extant and extinct species counts, while red numbers are extinct species. We follow PHYLACINE 1.2.1 for mammal ranges and species list of all extant and extinct mammals throughout the last 129,000 years and include prehistorically extinct (EP), historically extinct (EX) and extinct in the wild (EW) as extinct. Continental extirpations are counted as extinctions in the bottom panels. The figure only includes non-marine species (i.e., sea cows, whales, seals, and marine otters are excluded), and also excludes humans (*Homo* spp.) and island endemics.

**Table 1.** Extant and extinct terrestrial megaherbivores (mean adult body weight ≥1,000 kg) from the late Quaternary

| Extant – 11 species[a] | Extinct – 46 species |
|---|---|
| **Cetartiodactyla** | |
| Bovidae | Bovidae[f] |
| – *Bos gaurus*[b], VU | – *Bubalus palaeokerabau* |
| – *Bubalus arnee*[b], EN | – *Pelorovis antiquus* |
| Giraffidae | Camelidae[g] |
| – *Giraffa camelopardalis*, VU | – *Camelops hesternus* |
| Hippopotamidae | Hippopotamidae |
| – *Hippopotamus amphibius*, VU | – *Hexaprotodon* sp.[b] |
| **Cingulata** | |
| | Chlamyphoridae[h] |
| | – *Doedicurus clavicaudatus* |
| | – *Glyptodon clavipes* |
| | – *Glyptotherium floridanum*[i] |
| | – *Panochthus tuberculatus* |
| **Diprotodontia** | |
| | Diprotodontidae |
| | – *Diprotodon optatum* |
| **Litopterna**[c] | |
| | – *Macrauchenia patachonica* |
| | – *Xenorhinotherium bahiense* |
| **Notoungulata** | |
| | Toxodontidae |
| | – *Mixotoxodon larensis* |
| | – *Toxodon platensis* |
| | – *Trigodonops lopesi* |
| **Perissodactyla** | |
| Rhinocerotidae | Rhinocerotidae |
| – *Ceratotherium simum*, NT | – *Coelodonta antiquitatis* |
| – *Diceros bicornis*, CR | – *Elasmotherium sibiricum* |
| – *Rhinoceros sondaicus*, CR | – *Stephanorhinus hemitoechus* |
| – *Rhinoceros unicornis*, VU | – *Stephanorhinus kirchbergensis* |
| **Pilosa**[d] | |
| | Megalonychidae |
| | – *Megalonyx jeffersonii* |
| | – *Megistonyx oreobios* |
| | – *Meizonyx salvadorensis* |
| | – *Nohochichak xibalbahkah* |
| | Megatheriidae |
| | – *Eremotherium laurillardi* |
| | – *Megatherium americanum* |
| | – *Megatherium tarijense* |
| | Mylodontidae |
| | – *Glossotherium robustum* |
| | – *Glossotherium tropicorum* |
| | – *Glossotherium phoenesis* |
| | – *Lestodon armatus* |
| | – *Mylodon darwini* |
| | – *Mylodonopsis ibseni* |
| | – *Paramylodon harlani* |
| | – *Scelidodon chiliensis* |
| | – *Scelidotherium leptocephalum* |
| **Proboscidea** | |
| Elephantidae | Elephantidae |
| – *Elephas maximus*, EN | – *Mammuthus columbi* |
| – *Loxodonta africana*, EN | – *Mammuthus primigenius* |
| – *Loxodonta cyclotis*[e], CR | – *Palaeoloxodon antiquus*[j] |
| | – *Palaeoloxodon iolensis*[j] |
| | – *Palaeoloxodon mnaidriensis*[j] |
| | – *Palaeoloxodon namadicus*[j] |
| | – *Palaeoloxodon naumanii*[j] |

(*Continued*)

**Table 1.** (*Continued*)

| Extant – 11 species[a] | Extinct – 46 species |
|---|---|
| | Gomphotheriidae |
| | – *Cuvieronius hyodon* |
| | – *Notiomastodon platensis* |
| | Mammutidae |
| | – *Mammut americanum* |
| | Stegodontidae |
| | – *Stegodon orientalis*[k] |
| | – *Stegodon trigonocephalus* |

*Note:* Information from IUCN Red List (2023) for extant species and conservation status, and PHYLACINE 1.2.1 for extinct species (Faurby et al., 2018, 2020a), except as indicated in footnotes. IUCN Red List categories: CR, Critically Endangered; EN, Endangered; NT, Near Threatened; VU, Vulnerable. The table primarily follows the taxonomy in PHYLACINE 1.2.1 but deviates where the IUCN taxonomy has been updated since the latest PHYLACINE release, and where newer information is available for extinct species than PHYLACINE 1.2.1.
[a]All population sizes follow IUCN Red List (2023).
[b]Jukar et al. (2021).
[c]Body masses following Croft et al. (2020).
[d]Information from PHYLACINE 1.2.1 updated with McDonald (2023).
[e]*Loxodonta cyclotis* is split from *L. africana* in latest IUCN taxonomy.
[f]Various Late Pleistocene populations of other Bovini species had populations with average adult body masses ≥1,000 kg, for example, *Bison* and *Bos primigenius* (e.g., Saarinen et al., 2016).
[g]Multiple extinct llamas removed relative to PHYLACINE 1.2.1 due to updated body mass estimates.
[h]Family name following Delsuc et al. (2016).
[i]PHYLACINE 1.2.1 included *Glyptotherium floridanum* and *G. cylindricum*, but these taxa are now lumped (Zurita et al., 2018).
[j]All *Palaeoloxodon* species were listed as *Elephas* in PHYLACINE 1.2.1.
[k]Late Pleistocene *Stegodon* from India is often referred to a separate species, *S. namadicus* (Jukar et al., 2021).

megaherbivore size threshold, the giraffe (*Giraffa camelopardalis* s.l.) and two bovines. The extinct species include, for example, a variety of proboscideans and rhinoceroses, giant ground sloths and armadillos, rhinoceros-like toxodonts, and a giant marsupial. Many larger meso-herbivores (≥50 kg) also went extinct during this time including, for example, numerous equids, camelids, bovids, marsupials, ground sloths, and armadillos (Faurby et al., 2018). Additionally, numerous large predators ≥50 kg went extinct, including all of the remaining saber-toothed cats (*Smilodon* spp., *Homotherium latidens*), steppe lions (*Panthera atrox* and *P. spelaea*), dire wolf (*Aenocyon dirus*), several bears, and a large hypercarnivorous marsupial (*Thylacoleo carnifex*) (Faurby et al., 2018). Comparatively few smaller mammals went extinct during the late Quaternary, though non-insular excep-tions in the 10–49 kg range include a collection of small ungulates (e.g., *Antidorcas bondi*), mid-sized marsupials and monotremes (e.g., *Borungaboodie hatcheri*, *Megalibgwilia robustus*), armadillos (e.g., *Dasypus bellus*), canids (e.g., *Protocyon troglodytes*), and large monkeys (e.g., *Caipora bambuiorum*) (Faurby et al., 2018). Multiple large-scale extirpations also occurred among mammals ≥10 kg in the late Quaternary, with leopards (*Panthera pardus*) and dholes (*Cuon alpinus*) and several other species disappearing from Europe s.s. (Stuart, 2015; Taron et al., 2021). Extinctions in the 1–9 kg range were very few (Faurby et al., 2018; Smith et al., 2018). The fragmented nature of the fossil record limits our ability to accurately estimate extinction rates with new species being discovered (Stinnesbeck et al., 2017), shown to be invalid (Zurita et al., 2018), or identified in later time periods (Yang et al., 2019). Generally, however, these new findings show past extinction estimations to be too conservative. We also note the comparative ease of identifying extinctions in large mammals given their more robust bones, facilitating preservation. However, their relatively low population densities provide a coun-teracting effect. Due to its recency, the late-Quaternary inherently has a much more complete fossil record than earlier times, and its

peculiar pattern of size-biased extinctions, relative to the rest of the Cenozoic, is very robust (e.g., Stuart, 1991; Smith et al., 2018).

### Extinctions in non-mammalian animals

In addition to mammalian megafauna, certain other groups suffered substantial extinctions during this period. These are mostly large-sized (≥10 kg body mass) or, if smaller, relatively large for their phylogenetic group, or species with links to megafauna, such as large dung beetles and a large vampire bat (Galetti et al., 2018; Tello et al., 2021). Among birds, multiple and mostly large to very large scavenging birds from the Americas (Galetti et al., 2018; Jones et al., 2018) and the only known vulture from Australia (Mather et al., 2022) went extinct. Also affected were various giant flightless birds such as an Asiatic ostrich (*Pachystruthio anderssoni*) (Buffetaut, 2023), the Australian mihirung (*Genyornis newtoni*) (Demarchi et al., 2022), and a late-surviving terror-bird from southern South America (*Psilopterus* sp.) (Jones et al., 2018). Other more moderately large birds went extinct on the continents as well, including an endemic Californian turkey (*Meleagris californica*) and a large coot (Bochenski and Campbell, 2006; Alarcón-Muñoz et al., 2020). A large number of bird extinctions occurred on islands, biased toward larger and flightless species (Fromm and Meiri, 2021), for example, huge fowl on Fiji and New Caledonia (Sylviornithidae) (Worthy et al., 2016), and truly giant flightless elephant birds (Aepyornithiformes) and moas (Dinornithiformes) on Madagascar and New Zealand, respectively (e.g., Koch and Barnosky, 2006; Grealy et al., 2023). Among reptiles, a number of giant tortoises went extinct on the mainland during the Late Pleistocene with additional species on islands from this time onward (Rhodin et al., 2015). In addition, multiple crocodiles (Ristevski et al., 2023), large monitor lizards (*Varanus* spp.) (Hocknull et al., 2009), a large snake (*Wonambi naracoortensis*), and a giant armored lizard (Thorn et al., 2023) went extinct in Australasia (Palci et al., 2018), and an enormous freshwater turtle disappeared in Amazonia (Ferreira et al., 2024).

### Geographic patterns

The prehistoric late-Quaternary extinctions exhibit well-documented geographic contrasts, with moderate extinctions in the Afrotropics and Indomalaya, more severe extinctions in the Palaearctic, even more severe extinctions in the Nearctic and Neotropics, and near-total loss in Australasia (e.g., Martin, 1966, 1967; Koch and Barnosky, 2006; Stuart, 2015; Figure 1). Importantly, there is a strong size basis in these extinctions even in the lesser affected regions (Figure 1). Further, many island environments also experienced total to near-total extinctions of their larger native fauna within this period or later (Stuart, 2015). Sub-Saharan Africa is often presented as having an intact megafauna, but actually lost a number of species in the late Quaternary, including an elephant species (*Palaeoloxodon iolensis*), a giant buffalo (*Pelorovis antiquus*), various antelopes (e.g., *Rusingoryx atopocranion*) and a giant warthog (*Metridiochoerus compactus*) (Faith, 2014; Manthi et al., 2020; Kovarovic et al., 2021). Similarly, mainland southern Asia also lost an elephant (*Palaeoloxodon namadicus*), another proboscidean (*Stegodon orientalis*), a hippopotamus (*Hexaprotodon sivalensis*), an equid (*Equus namadicus*), several orangutans (*Pongo* spp.), and a giant tapir (*Tapirus augustus*), while the Indian aurochs (*Bos primigenius namadicus*) only survived in domesticated form (*B. p. indicus*) (Bacon et al., 2015; Jukar et al., 2021). Northern Eurasia and North Africa also lost substantial numbers of megafauna during

the Late Pleistocene and through the Holocene, for example, multiple species of elephant, rhinoceros (including the giant *Elasmotherium sibiricum*) and giant deer (e.g., *Megaloceros giganteus*), with the aurochs (*B. primigenius primigenius*) only surviving in domesticated form (*B. p. taurus*) (Faith, 2014; Stuart, 2015; van der Plicht et al., 2015; Kosintsev et al., 2019). North and South America experienced more severe losses, including all proboscideans, ground sloths, glyptodonts, the endemic ungulate orders (Notoungulata, Litopterna), equids, and all but two camelids (e.g., Stuart, 2015). Australia lost almost all of its megafauna, with the complete extinction of all terrestrial species ≥50 kg, which included multiple species of short-faced kangaroo and several giant wombat relatives (Stuart, 2015). Extinctions of smaller megafauna occurred on essentially all major island systems at different times over the Late Pleistocene to the Late Holocene (e.g., Koch and Barnosky, 2006; Slavenko et al., 2016; Andermann et al., 2020; Fromm and Meiri, 2021).

Megafauna losses occurred in a broad variety of biomes (Figure 2). Extinctions from cold, northerly biomes are well-documented, notably woolly mammoth (*Mammuthus primigenius*) and woolly rhinoceros (*Coelodonta antiquitatis*), alongside the extirpation of muskoxen (*Ovibos moschatus*) from Eurasia (e.g., Lorenzen et al., 2011; Stuart, 2015). However, these species represent only very few of 134 known late-Quaternary extinctions among terrestrial mammal species ≥100 kg average adult body mass (Figure 1), hereunder 14 island endemics. The remainder occurred in tropical to temperate climates in ecosystems ranging from dense forests to open woodlands and savannahs to grasslands and deserts, and extinction occurred with similar severity across biomes (Figure 2). Importantly, a large proportion of the extinct species were clearly generalist in terms of climate, habitat, and diet (e.g., Price, 2008; França et al., 2015; van Asperen and Kahlke, 2015; Rivals et al., 2019). Even woolly mammoth and woolly rhinoceros occasionally occurred in temperate or boreal settings during the Late Pleistocene, for example in Iberia (Álvarez-Lao and García, 2012) and north-eastern China (Ma et al., 2021).

### Temporal patterns

Temporal staggering is a defining feature of the late-Quaternary megafauna extinctions, as already noted by Martin (1967), with extinctions concentrated in different time windows in different areas, extending from ~50,000 years ago until well into the Holocene, and often spread across thousands of years even within a given region. The extinctions of megafauna in Australia and New Guinea primarily occurred between 60–40 kya, whereas extinctions in the Americas began roughly 20–15 kya (e.g., Stuart, 2015) and continued until as late as 7–5 kya (Prado et al., 2015; Murchie et al., 2021). Extinctions in Eurasia occurred in different regions at different times during this 60–5 ky span (Stuart, 2015). Megafauna extinctions on islands exhibited similar staggering, albeit tending to happen later, from the end of the Pleistocene onward. Examples include Japan (25,000–16,000 ya) (Iwase et al., 2012), the California Channel Islands (~13,000 ya) (Rick et al., 2012), Sardinia (~7,500 ya) (Benzi et al., 2007), the Antilles (~4,000 ya) (Steadman et al., 2005), New Caledonia (~3,000 ya) (Anderson et al., 2010), Madagascar (~1,000 ya) (Hansford et al., 2021), and New Zealand (~600 ya) (Perry et al., 2014). A notable example of survival deep into the Holocene is the widespread survival of woolly mammoth (*Mammuthus primigenius*) until the Middle Holocene in northern parts of continental Siberia and North America (Murchie et al., 2021; Wang et al., 2021). Similar late survival in Siberia is also noted for woolly rhinoceros (*Coelodonta antiquitatis*), steppe bison (*Bison*

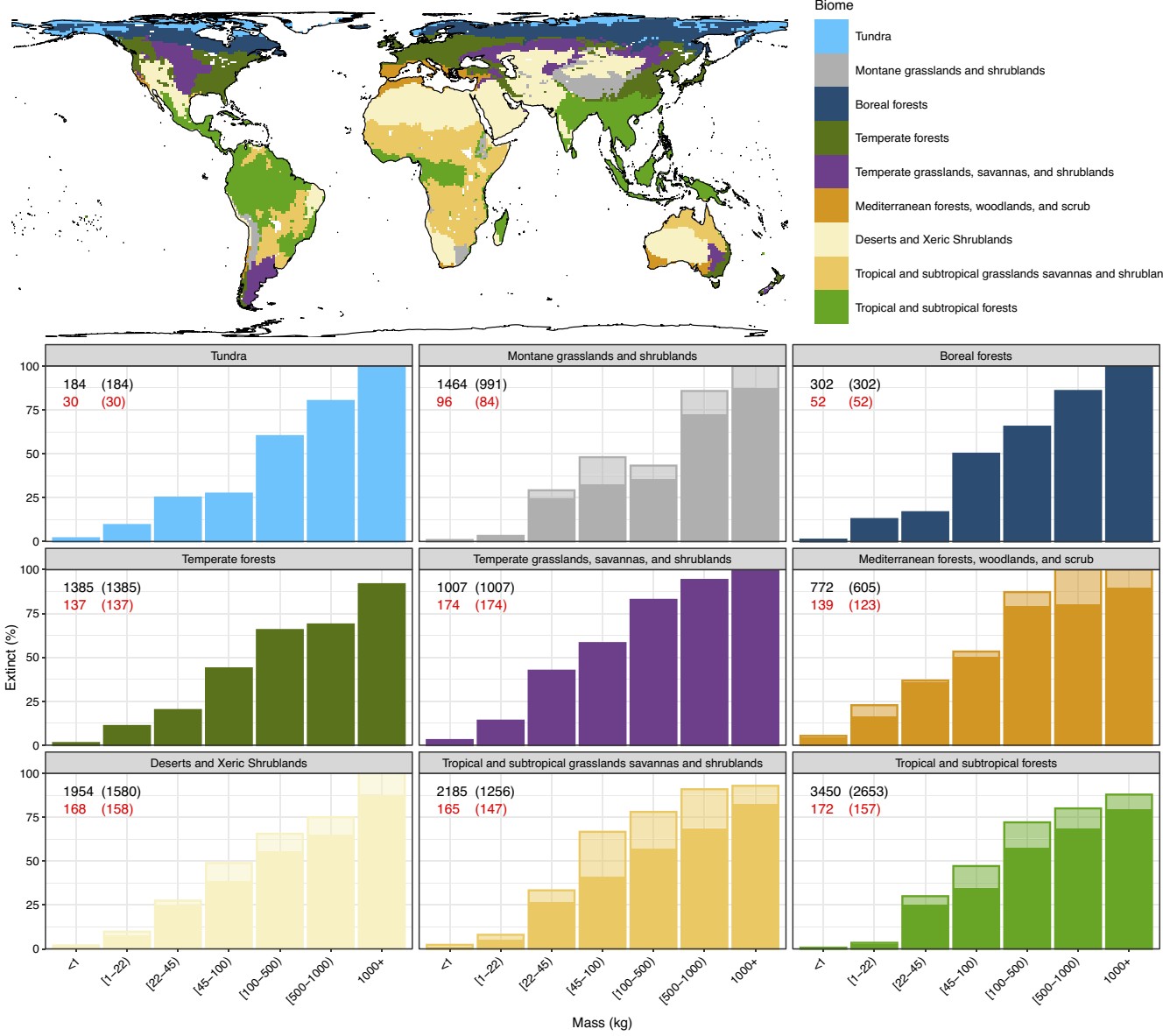

**Figure 2.** Late-Quaternary mammal extinction as a function of body size and current biome. Black numbers are total late-Quaternary extant and extinct species counts, while red numbers are extinct species. The shaded part of the bars is the extinction fraction if we exclude Africa from the analysis – to illustrate how much its relatively numerous remaining megafauna influence the patterns. We follow PHYLACINE 1.2.1 for present-natural and current species ranges and the species list of all extant and extinct mammals throughout the last 129,000 years, counting species as extinct in a biome if it no longer occurs there. The figure only includes non-marine species (i.e., with sea cows, whales, seals, and marine otters excluded), and also excludes humans (*Homo* spp.) and island endemics. For biomes, we follow the Terrestrial Ecoregions of the World by the World Wildlife Fund.

*priscus*), giant deer (*Megaloceros giganteus*), and the extant muskox (*Ovibos moschatus*) (van der Plicht et al., 2015; Plasteeva et al., 2020; Wang et al., 2021). East Asian examples include the non-caballine horse *Equus ovodovi*, a temperate water buffalo (*Bubalus mephistopheles*), and the ostrich *Pachystruthio anderssoni* (Janz et al., 2009; Turvey et al., 2018; Cai et al., 2022). Other examples include European wild ass (*Equus hydruntinus*) (Crees and Turvey, 2014), giant buffalo (*Pelorovis antiquus*) in Africa (Faith, 2014), the extant wild horse (*Equus ferus*) in North America (Murchie et al., 2021), and various megafauna species in South America, for example the proboscidean *Notiomastodon platensis* (Dantas et al., 2022), the large ground sloth (*Scelidotherium leptocephalum*), and a giant armadillo *Eutatus seguini* (Prado et al., 2015). Some species

that were more widespread in the Pleistocene or Early Holocene had their last stands in historic times, for example Steller's sea cow (*Hydrodamalis gigas*) and a large, near-flightless cormorant (*Urile perspicillatus*), both of which ranged across the North Pacific during the Pleistocene and earlier Holocene (Crerar et al., 2014; Watanabe et al., 2018). Tasmania held the last thylacines (*Thylacinus cynocephalus*) until European colonization, and still holds the last devils (*Sarcophilus harrisii*) and flightless nativehens (*Tribonyx mortierii*), with the disappearance of all three species in mainland Australia coinciding with the human introduction of dingoes (*Canis lupus dingo*) (Letnic et al., 2012).

A few species of megafauna went extinct in their originally wild form in the Late Holocene or recorded history, but survive as

domesticates, such as cattle (*B. primigenius taurus* and *B. primigenius indicus*) (Ajmone-Marsan et al., 2010), and one or both species of domestic camels (*Camelus bactrianus* and *C. dromedarius*) (Fitak et al., 2020). The historical record also documents many instances of Holocene range declines in extant megafauna species, for example, in the Middle East (Tsahar et al., 2009; Bar-Oz et al., 2011), China (Turvey et al., 2017, 2018; Teng et al., 2020), and Europe (Crees et al., 2016). A recent study on 139 species of extant terrestrial megafauna mammals from all continents (except Antarctica) using analyses of current genomes to estimate past demographics shows strong prehistoric population declines in >90% of these species and no significant increases (Bergman et al., 2023; Figure 3), in line with earlier studies on specific megafauna groups, for example rhinoceroses (Liu et al., 2021) and elephants (Palkopoulou et al., 2018). According to breakpoint analyses, these declines started at 76,000–32,000 years ago, at different times in different realms, and their severity increased with body mass (Bergman et al., 2023). Furthermore, many of the surviving megafauna species exhibit body size declines across the Late Pleistocene and Holocene. American bison (*Bison bison*), for example, had an average body size in the Late Pleistocene that was 37% greater than today (Martin et al., 2018). Other examples include jaguars (*Panthera onca*) in North and South America (Srigyan et al., 2023), coyotes (*Canis latrans*) in North America (Meachen et al., 2014), European brown bears (*Ursus arctos*) (Marciszak et al., 2015), European wild horse (*E. ferus*) (Forsten, 1993), and various kangaroos and other larger Australian marsupials (Marshall and Corruccini, 1978). Hence, we can generalize the prehistoric dynamics of surviving megafauna as a series of size-biased declines similar to the extinction patterns.

Just as historic, Holocene, and Late Pleistocene extinction patterns can be difficult to separate, the Late Pleistocene extinctions also grade into earlier losses in some instances. There is evidence that megafauna extinctions without replacement and with a tendency towards body-size downgrading started somewhat earlier than the Late Pleistocene in some regions, specifically Africa and Eurasia (e.g., Martin, 1966). The large carnivore guild in Sub-Saharan Africa underwent a drastic simplification including the loss of all machairodont cats already in the Early Pleistocene (Geraads, 2018; Faurby et al., 2020b). Machairodont cats also went extinct or became rare (*Homotherium latidens*) in Eurasia from the early or middle Middle Pleistocene onward (Antón et al., 2005). Megaherbivores declined in diversity in Africa from the Early Pleistocene onward (Faith, 2014; Bibi and Cantalapiedra, 2023). More generally, a variety of large herbivores also went extinct before the Late Pleistocene seemingly without replacement in both Africa and southern Asia, as already noted by Martin (1966, 1967), although this is not always clearcut to discern. Potential examples include the proboscidean *Deinotherium*, chalicotheres, robust giraffes (*Sivatherium giganteum*), the giant geleda *Theropithecus oswaldi*, and the giant terrestrial ape *Gigantopithecus blacki* (Surovell et al., 2005; Cerling et al., 2015; Yan et al., 2016; Geraads, 2018; Zhang et al., 2024). A large beaver (*Trogontherium cuvieri*) also disappears from most of Eurasia at the same time (Yang et al., 2019). In Sub-Saharan Africa, these losses were strong enough to make large-herbivore faunas from 700,000 years ago and earlier functionally non-analog to extant faunas due to a greater richness of non-ruminants and megaherbivores (Faith et al., 2018; also see Bibi and Cantalapiedra, 2023).

### Other potential patterns

While there is evidence of co-extinctions in non-megafauna species dependent on large vertebrates (Galetti et al., 2018), the possibility of co-extinction among megafauna merits consideration as well (e.g., Owen-Smith, 1987). Extinction of the largest herbivores capable of preventing the dominance by woody plants or coarse grasses would have been to the disadvantage of smaller herbivores due to less nutrient-rich and spatially diverse vegetation (Owen-Smith, 1987; also cf. Trepel et al., 2024). In line with that possibility, several small, grassland-associated gazelles disappear in Africa (Faith, 2014). Multiple pronghorns, including small and very small species, similarly went extinct in North America (e.g., Bravo-Cuevas et al., 2013), and the extant saiga (*Saiga tatarica*), a small antelope associated with open habitats, disappears from much of its Holarctic range (Jürgensen et al., 2017). Small, ruminant grazers require high-quality grasses and forbs, the maintenance of which often requires larger herbivores to remove more low-quality vegetation, as seen in the relationship between Thomson's gazelle (*Eudorcas thomsonii*) and larger grazers like zebra (*Equus quagga*) and wildebeest (*Connochaetes taurinus*) in East Africa (Bell, 1971; Anderson et al., 2024).

### Extinction drivers

The debate on the cause or causes of the late-Quaternary extinctions has been ongoing for over 200 years, but with greater rigor and focus from the 1960s onward (Martin, 1967; Koch and Barnosky, 2006). Though most current work accepts at least a contributory role for modern humans, the topic remains controversial. Competing alternative theories include an extra-terrestrial impact (reviewed and rejected by Holliday et al., 2023) and, more credibly, climate change.

To evaluate support for different drivers, there are a number of characteristics of the late-Quaternary extinctions that any hypothesis needs to account for. Firstly, the late-Quaternary extinction was a global event (Figure 1), largely constrained to the Last Glaciation and the Holocene, and unique for the whole of the Cenozoic (Smith et al., 2018; Figure 4). It occurred in all climate zones, with the majority of extinctions happening among temperate to tropical species (Figure 2). The late-Quaternary extinctions were extremely size-selective, with high and positively size-dependent extinction rates among large terrestrial vertebrates and very limited extinctions within the same time frame in smaller-sized terrestrial animals, marine vertebrates of any size, or plants (e.g., Smith et al., 2018). Further, the extinct species come from a large number of mammal families and orders and extend to a variety of only distantly related birds and reptiles. Additionally, severe extinctions penetrated to smaller animals on islands, but still with a bias toward the largest species there (e.g., Fromm and Meiri, 2021). Finally, the extinctions were concentrated in different time windows in different continents, regions and islands, and extended from, for example, as early as ~50,000 years ago to the Middle and even late Holocene (e.g., Crees and Turvey, 2014; van der Plicht et al., 2015; Andermann et al., 2020; Cai et al., 2022). These patterns mean that we can immediately discard explanations that do not have global scope or that hinge on a particular event such as a late-glacial extra-terrestrial impact or the loss of a specific ecosystem type like the mammoth steppe. Importantly, while detailed studies of range dynamics in single extinct species are valuable, explanatory models are only tenable if we can generalize them to the broader late-Quaternary extinct event. This leaves two broad potential drivers for serious consideration, namely the spread and cultural evolution of *Homo sapiens* and climate change during the late Quaternary. The possibility also exists for interaction between the two (e.g., Koch and Barnosky,

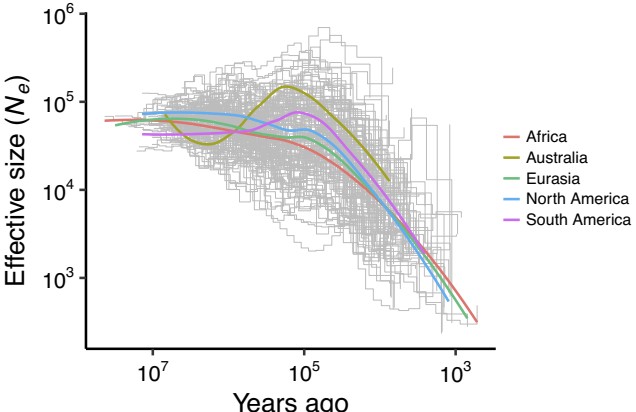

**Figure 3.** Genomic analyses show that surviving large mammals experienced strong population declines across the late Quaternary, in parallel to the global extinctions of many megafauna species. Effective population size dynamics were inferred from the whole genome nucleotide diversity of 139 terrestrial mammals (all >10 kg body mass) using the Pairwise Sequentially Markovian Coalescent method, adapted from Bergman et al. (2023). Each gray step line represents a population size trajectory of a single species with the average population trend for each continent depicted in color. Both axes are log₁₀-transformed.

2006). Here, we first discuss the evidence with respect to a climatic cause, and then with respect to *Homo sapiens*.

### Climatic causation

The Late Pleistocene saw intense climatic shifts, which are often implicated as drivers of megafauna extinctions (e.g., Lorenzen et al., 2011; Faith, 2014; Stewart et al., 2021). Various episodes of climatic stress or fast change have been proposed as causing the megafauna extinctions, for example: rapid Dansgaard-Oeschger warming events between 56,000–12,000 ya (Cooper et al., 2015), regional longer-term climatic drying within the middle part of the last glaciation (Hocknull et al., 2020), but also even the relative stability of the Holocene itself (Mann et al., 2019).

A fundamental challenge for these explanations is that earlier severe climate instability did not lead to a similar pattern of extinction. Many earlier glacial cycles occurred during the Pleistocene, but only the most recent is associated with widespread and highly size-selective megafauna extinctions. Severe regional extinctions did occur earlier in the Pleistocene and were clearly climate-linked, but these affected a variety of non-megafauna organism groups, for example woody plants in Europe (Svenning, 2003; Magri et al., 2017; Martinetto et al., 2017) and Australia (Mooney et al., 2017), and mollusks in the Atlantic and the Mediterranean (Stanley and Campbell, 1981; Raffi et al., 1985). These earlier extinctions did not lead to depauperate megafaunas and form a strong contrast to the megafauna-selective extinctions of the late Quaternary. This contrast extends even further back; earlier in the Cenozoic there were also major climate changes and associated extinctions, but these were also not size-selective and did not lead to depauperate megafaunas (Smith et al., 2018). An illustrative case concerns proboscideans in Europe and North America. Except for the last 0.01 million years, both continents have continuously harbored proboscideans throughout the last 18 and 16 million years, respectively, despite intense climate fluctuations (e.g., Fox and Fisher, 2004; Lucas and Morgan, 2005; Haiduc et al., 2018; von Koenigswald et al., 2023). The contrast between these earlier non-size-selective

extinction patterns and the strongly size-selective extinctions in the late Quaternary is a strong argument against a climatic causation.

An important consideration is if the last glacial cycle was somehow more severe than earlier ones, as this might then potentially explain its unique megafauna losses. There is, in fact, little support for such a scenario. Overall, Earth has had the same climate regime during the last ~1 million years, since the shift to deep, long (~100 kyr) glacial-interglacial cycles with the Mid-Pleistocene Climate Transition (Clark et al., 2006; Herbert, 2023). Maximal Pleistocene ice sheet cover in the Northern Hemisphere was attained during multiple Middle Pleistocene glaciations rather than during the last glacial cycle (Batchelor et al., 2019). Similarly, multiple glacial cycles during the Middle Pleistocene and late Early Pleistocene led to more severe cooling and greater vegetation change (Margari et al., 2023). Fast-paced, extreme climatic shifts similar to Heinrich events of the Last Glaciation are also documented from the earlier glacial cycles, having occurred since the Mid-Pleistocene Climate Transition (Naafs et al., 2013). Further, strong millennial-scale climate variability similar to Dansgaard-Oeschger events of the last glaciation has been typical of glacial climates for at least the last 1.5 million years (Hodell et al., 2023; Margari et al., 2023). Importantly, such frequent and persistent millennial climate instability prior to the Late Pleistocene has been shown to have had pronounced impacts on terrestrial ecosystems (e.g., vegetation) even within glacial refugial areas (Wilson et al., 2021), but nevertheless did not elicit selective megafauna extinction episodes. Altogether, Quaternary climate history does not provide any obvious mechanism for the unique extinction pattern of the Late Pleistocene and Holocene.

On a global scale, megafauna extinction severity only poorly links to the severity of glacial maximum-present climate shift, with severe extinctions in many relatively stable regions such as California, southern Australia, and the pampas region of South America (Sandom et al., 2014a; Lemoine et al., 2023). The continual climate changes throughout the late Quaternary mean that extinction-climate links may easily appear to be present in any restricted spatiotemporal window, that is, if the longer-term and broader geographic contexts are not considered. Further, strong climate change is predicted to elicit range and abundance responses in most species. In multiple cases, apparent regional population and community responses to climate have been suggested to support climate-driven extinction (e.g., Lorenzen et al., 2011; Cooper et al., 2015; Stewart et al., 2021; Wang et al., 2021). However, such dynamics may reflect normal range responses to climate, as seen in numerous surviving species in response to the severe climatic changes of the period (e.g., Sommer et al., 2014; Cooper et al., 2015). Further, other studies show contrasting patterns. For example, a sedimentary ancient DNA (sedaDNA) study from the Yukon shows strong megafauna decline between 21 and 14,500 years ago, prior to the loss of the mammoth steppe biome and the Younger Dryas (Murchie et al., 2021). In addition, as already mentioned, many extinct megafauna species have last occurrences in the Early or even Middle Holocene, that is, during the relatively stable climate of the Holocene, meaning that a climatic cause for their extinction is unlikely given their previous survival through numerous, massive climatic shifts throughout the Pleistocene, including long and warm interglacial periods.

An increasing number of studies look at local and regional dynamics in the overall abundance of large herbivores at high spatiotemporal resolution using dung-associated fungal spores. Many of these are able to pinpoint declines to timeframes where the climate was stable, for example, North America ~14–

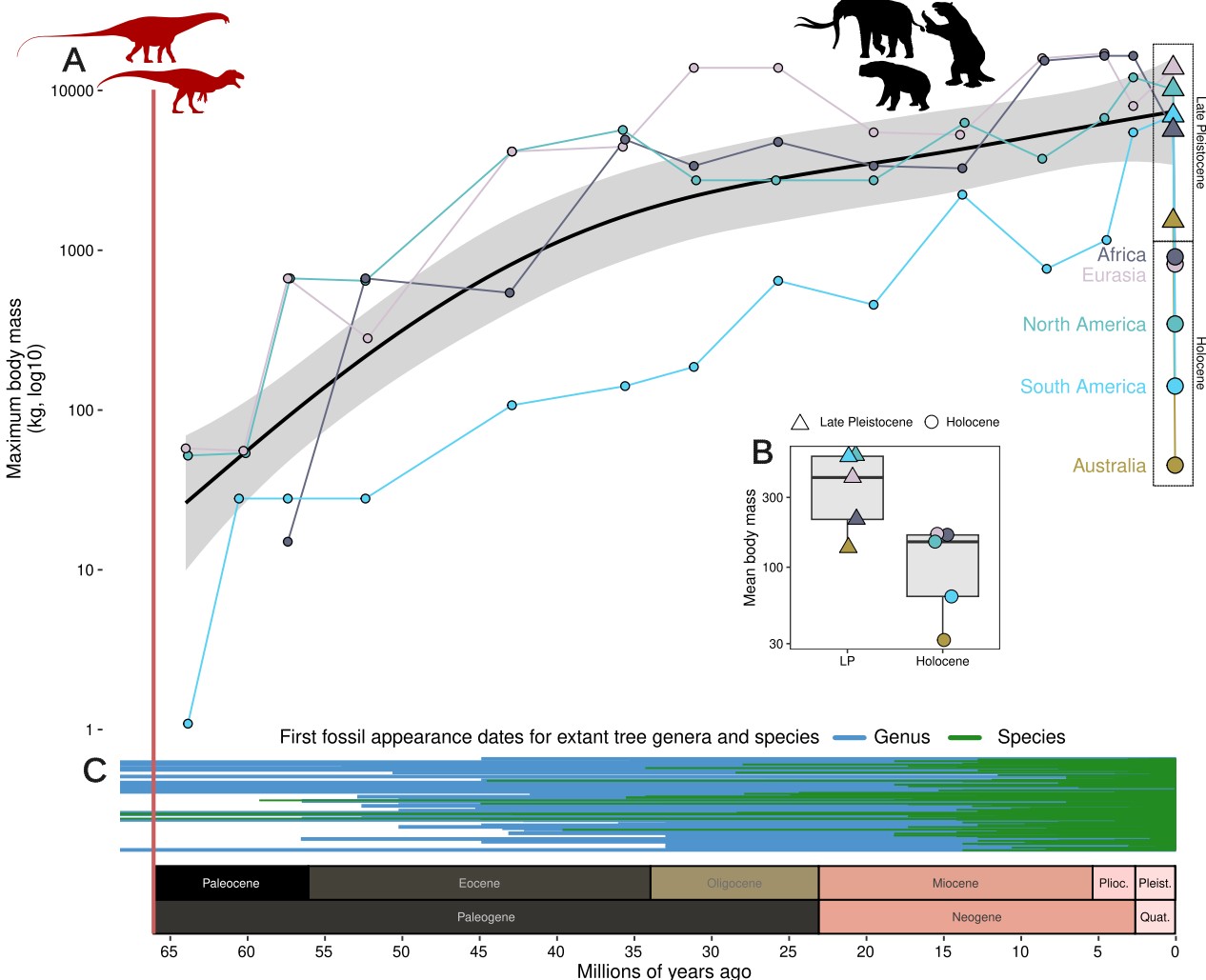

**Figure 4.** Mammalian body masses through the Cenozoic. (A) Maximum mammalian body mass (kg, log10 scale) increased steadily following the end-Mesozoic mass extinction (vertical red line) until the Late Pleistocene, but then declines precipitously. The *X*-axis indicates millions of years ago and is shared across panels A and C with geologic periods and epochs illustrated above axis text. Black line with gray error indicates Generalized Additive Model estimate across maximum values for four continents (data unavailable for Australia). Colored lines indicate the separate continental maximum body mass through time. Declines from the Late Pleistocene to the Holocene are highlighted with dashed boxes. (B) Mean body mass per continent during the Late Pleistocene and Holocene, after the megafauna extinctions. Data for A and B are from Smith et al. (2018). (C) Megafauna was a pervasive presence during the evolution of modern-day terrestrial biota, here illustrated with first appearance dates of extant tree genera (blue) and species (green), shown as lines from the first fossil record to today (data from Paleodb, accessed 2023).

13,000 years ago, prior to the Younger Dryas cooling (Gill et al., 2009; Halligan et al., 2016; O'Keefe et al., 2023), and 41,000 years ago in Australia at a time of no substantial climate change (Rule et al., 2012; Adeleye et al., 2023). In southeastern Brazil, the faecal spore decline spans from 15 to 11,000 years ago, overlapping a climatic wetting episode (Raczka et al., 2018), and a similar coincidence is reported for a site at 3,000 m in the Peruvian Andes, between 16,800 and 15,800 years ago (Rozas-Davila et al., 2016). However, these climate episodes were no more severe than the numerous others that occurred during the late Quaternary. Further, a 4,000 m site, also in the Peruvian Andes, finds that the decline occurs at 13–12,300 years ago and is not linked to any obvious climatic event (Rozas-Davila et al., 2023). In eastern North America strong, chronic spore declines occur during a relatively mild climate interval prior to the Younger Dryas, showing that this cooling episode cannot be the cause of the overall megafauna decline (e.g., Gill et al., 2009). Emerging sedaDNA studies are from high northern latitudes, where strong climate imprints on biotic

dynamics are expected (Murchie et al., 2021; Wang et al., 2021). They report megafauna-climate relations, but with some megafauna survival deep into the relatively stable Holocene (Murchie et al., 2021; Wang et al., 2021) and sometimes with major declines prior to shifts in climate and vegetation (Murchie et al., 2021). Altogether, the detailed spatiotemporal resolution offered by the increasing number of fungal and sedaDNA studies does not support a strong role of climate in the chronic declines in megafauna.

Range modeling has been used to test if extinction is explainable from reduced climatic suitability through the late Quaternary, for example estimating strong reductions in climatically suitable areas for woolly mammoth (Fordham et al., 2022). However, such models estimate the realized niche, that is, the conditions under which the species occurred during the period where the calibration data come from, including pressure from humans, and hence might potentially estimate climatic refugia from Paleolithic hunting (cf. Pitulko et al., 2016). Since most of the late-Quaternary extinctions affected temperate to tropical species (Figure 2), late- and post-glacial

declines in climatic suitability would not be expected for most species. Accordingly, range models for many extinct megafauna species have indicated stable, rising or at least large remaining areas of climatically suitable conditions into the Holocene (Martinez-Meyer et al., 2004; Varela et al., 2010; Lima-Ribeiro et al., 2013; Villavicencio et al., 2019). These results can be seen as having been empirically validated by the successful reintroductions of species that suffered prehistoric regional extirpations, notably horse (*Equus ferus*) in North America and muskox (*Ovibos moschatus*) in Eurasia (Lundgren et al., 2018).

An important piece of evidence that counts against any climate explanation for the late-Quaternary extinctions concerns the evidence for broad climatic tolerances, broad dietary niches, and persistent food availability for many of the extinct megafauna species. Large carnivores exemplify these patterns well, with broad climatic ranges and dietary flexibility, for example scimitar cats (*Homotherium latidens*), which in the Late Pleistocene in North America spanned from Alaska to Texas and had a diet varying from generalist foraging on large herbivores to more focus on juvenile large grazers, and short-faced bears (*Arctodus simus*) with a similar distribution and a foraging strategy including various ungulates as well as plants (Bocherens, 2015; DeSantis et al., 2021; Smith et al., 2022). Occurrence across widely varying climatic conditions and high levels of dietary flexibility were common for late-Quaternary herbivores (e.g., Price, 2008; França et al., 2015; van Asperen and Kahlke, 2015). Tooth-wear and isotope studies indicate high dietary variability among conspecific populations and individuals, from mainly grazing to mainly browsing, in extinct proboscideans from Eurasia and the Americas such as *Mammuthus columbi, Palaeoloxodon antiquus, Notiomastodon platensis* and *Cuvieronius hyodon* (González-Guarda et al., 2018; Haiduc et al., 2018; Rivals et al., 2019; Smith and DeSantis, 2020; Dantas et al., 2022). For example, the South American proboscidean *Notiomastodon platensis* occurred in multiple ecosystem types and varied in diet from grazer to mixed-feeder or browser depending on locality (e.g., Asevedo et al., 2012; González-Guarda et al., 2018; Pérez-Crespo et al., 2020; Asevedo et al., 2021; Dantas et al., 2022). The American mastodon (*Mammut americanum*) exhibited a narrower signal as a consistent browser, but consumed a rich diversity of trees, shrubs, lianas, vines and herbs in Florida to spruce (*Picea*) and sedge-swamp plants in northern areas (Newsom and Mihlbachler, 2006; Green et al., 2017; Birks et al., 2019), leading to the conclusion that regional populations were capable of maintaining their dietary niche despite climate change (Green et al., 2017). Furthermore, studies of mastodon tusks indicate that a decreasing age in maturation over the last 1,000 years before their extinction, indicating that they were not experiencing nutritional stress (Fisher, 2009), as was also concluded based on the rich diversity of plants – most still regionally present – in their diet close to their extinction (Newsom and Mihlbachler, 2006). Similarly, toothwear studies in extinct large carnivores also do not indicate food shortage toward their extinction in the terminal Pleistocene (DeSantis et al., 2012).

Altogether, while it is clear that megafauna populations responded to the climate dynamics of the late Quaternary, like numerous other species, the combined evidence strongly challenges climate-based causal explanations for the late-Quaternary megafauna losses (Table 2).

## Human causation

In contrast to the substantial arguments against climate as a major cause of the late-Quaternary megafauna extinctions, their linkage to the rise and expansion of behaviourally modern humans during the Late Pleistocene and Holocene is well supported by many types of evidence, offering an explanation for why the global loss of megafauna, unique for the whole Cenozoic, happened at this time (Smith et al., 2018; Figure 4), as the emergence and global dispersal of *H. sapiens* was itself a unique event in Earth history, given our species' exceptional capabilities (e.g., Ellis, 2015). Notably, in areas with no pre-*sapiens* hominins, well-dated extinctions always occur around or after colonization by *H. sapiens*; additionally, extinction severity is strongly linked to human biogeography, with severe extinctions where *H. sapiens* was most novel (i.e., the first hominin present) and more moderate extinctions in areas of long-term human evolution (i.e., Africa and southern Asia), all discussed further below. Hence, the explanation most consistent with the extinction pattern is that expanding modern humans exploited megafauna at levels that were unsustainable in the short- or long-term.

There is widespread evidence that Late Pleistocene *Homo sapiens* populations targeted large game and were sophisticated, efficient megafauna hunters in the varied environments they colonized, for example: mammoths in northern Siberia as early as 45,000 years ago (Pitulko et al., 2016), diverse large mammals at the initial colonization of Europe north of the Alps at the same time (Smith et al., 2024), and a diversity of gomphotheres, giant ground sloths, giant armadillos, equids, bears, cervids, and camelids across South America (Bampi et al., 2022). In fact, numerous megafauna kill sites for a large number of extinct species exist in Africa (Kovarovic et al., 2021), Eurasia (Shipman, 2015; Pitulko et al., 2016), North America (Sanchez et al., 2014; Waters et al., 2015), and South America (Bampi et al., 2022). Kill sites are so far missing from the limited record for Australia and New Guinea (Hocknull et al., 2020). Many of the extinct megaherbivores (Table 1) likely had no predators as adults and are expected to have been vulnerable to the use of projectiles, traps, and fire (Agam and Barkai, 2018). Moreover, severe size-biased extinctions are also predicted from mechanistic modeling of human hunting and impacts on prey populations due to well-established general relationships of demographic characteristics to body mass (Alroy, 2001). Apart from hunting, other mechanisms of human-mediated megafauna extinctions have also been considered, notably habitat alteration, introduced predators (such as dogs), and introduced pathogens (Koch and Barnosky, 2006). However, these mechanisms cannot explain the size-selectivity, nor the timing and generality of the extinctions (Koch and Barnosky, 2006). Further, as discussed earlier, co-extinctions - rather than direct human persecution - help explain the extinctions that did occur among smaller species, clearly playing a role for directly megafauna-dependent species, but potentially also more widely. Notably, extinctions among various moderately sized species may potentially be linked to the disappearance of many or all of the largest herbivores (Owen-Smith, 1987).

Human biogeography provides a logical explanation for both the striking global spatiotemporal patterns in the late-Quaternary megafauna extinctions and for their severity. The lower severity and longer time frame of the extinctions in Africa and southern Asia matches the core areas of evolution of *Homo sapiens* as well as that of hominins broadly, as already noted by Martin (1966, 1967). A number of macroecological analyses have shown that the expansion of *H. sapiens* in the context of the broader human biogeographic history provides high statistical explanatory power for the global spatial and spatiotemporal patterns in the megafauna extinctions, whereas climate offers little such explanatory power (Sandom et al., 2014a; Bartlett et al., 2016; Araujo et al., 2017; Andermann et al.,

**Table 2.** Major arguments against a climatic causation of the prehistoric late-Quaternary megafauna extinctions and corresponding arguments for a human-caused explanation

| | Argument against climate-driven megafauna extinction | Argument for human-driven megafauna extinction |
|---|---|---|
| I | Despite a ~1 my pattern of ~100 kyr severe glacial cycles, with numerous rapid climatic events, widespread megafauna extinction without replacement only occurs within the last ~50 ky. Further, extinctions do not exhibit consistent overlap with climatically stressful periods within the latter time frame | The expansion of *Homo sapiens* out of Africa beginning ~100 kya is a unique event that had not yet occurred during previous glacial cycles, representing a novel factor in extinction dynamics. *H. sapiens* had colonized southern Eurasia and Australia by ~50 kya when megafauna extinctions began to accelerate. |
| II | The strong bias of the late–Quaternary extinctions toward larger terrestrial vertebrates is unique for the whole Cenozoic. There were strong climate–linked extinctions earlier in the Quaternary and Cenozoic, but these affected a broad range of taxa (e.g., plants and marine invertebrates) and were not associated with the selective loss of large–bodied vertebrates | Humans (*Homo* spp.) preferentially and efficiently hunted large game. Large species are more easily driven to extinction through increased adult mortality from hunting, as gestation and maturation periods are long and brood size is small. Within the Upper Paleolithic (~50–12 ky) modern humans (*H. sapiens*) attained far higher abundance than earlier hominins, consistent with their much stronger ecological effects |
| III | Spatiotemporal patterns in climate change during the late Quaternary cannot adequately explain the global pattern of extinction. Climate change over the past ~50 ky varied among continents, but in multiple cases matches poorly with extinction dates or severity. For example, Africa and South America experienced similar levels of climatic change but had vastly different extinction severity. Further, even relatively stable areas often experienced severe extinctions | Spatiotemporal patterns in late–Quaternary extinctions broadly correspond to *H. sapiens*' global colonization pattern, with extinctions occurring at or after arrival. The anomalously low extinction severity in Africa and southern Asia coincides with the longest occupation of hominins and is consistent with co–adaptation of the fauna to human predation, potentially combined with earlier losses of sensitive taxa and stronger suppression of human populations by pathogens in these long–occupied regions |
| IV | While range contractions are expected and observed for cold–adapted species in relation to Holocene warming, similar processes cannot explain extinctions among ecologically plastic species or among the much higher number of species associated with temperate or tropical climates | Human populations were likely greater in temperate and tropical regions than in colder regions. Hence, regions of less climatic stress or ecological plasticity in megafauna would not confer greater protection against human pressure |
| V | For scattered locations across Earth, megafauna declines have been pinpointed very precisely in time using the abundance of fungal spores that are specific to megafauna dung or using sedimentary ancient DNA (sedaDNA). Final declines are often not associated with periods of particularly high climatic stress or instability | Final megafauna declines, as indicated by fungal spores or sedaDNA, always occur close to or after modern humans became present in the broader region |

*Note:* The table summarizes the arguments outlined in the text, where the supporting references are provided.

2020; Lemoine et al., 2023). The same is true for severe population declines in numerous extant megafauna species over the last ~50,000 years, estimated based on current genomes (Bergman et al., 2023). When analyzing human and climatic predictors together, these global analyses all identify a dominant role for human biogeography, with climate providing a small (Sandom et al., 2014a; Bartlett et al., 2016) or negligible contribution (Araujo et al., 2017; Andermann et al., 2020; Bergman et al., 2023; Lemoine et al., 2023). Hence, the global analyses strongly support human causation, and even indicate little or no interaction with climate change, neither in the severity of extinction and decline nor in the timing. Further, the detailed spatiotemporal resolution offered by the increasing number of fungal and sedaDNA studies consistently shows that final megafauna declines occur close to or after the arrival of *H. sapiens*, for example in the arctic (Murchie et al., 2021; Wang et al., 2021), north-eastern (Gill et al., 2009), south-eastern (Halligan et al., 2016), and south-western North America (O'Keefe et al., 2023), eastern South America (Raczka et al., 2018) and the Andes (Rozas-Davila et al., 2016, 2023), and northern (Rule et al., 2012) and southern Australia (Adeleye et al., 2023). Also, severe megafauna losses in areas with relatively mild, stable climates are consistent with human causation, as human populations were likely greater in temperate and tropical regions than in colder regions (e.g., Ordonez and Riede, 2022). Hence, less climatic stress or ecological plasticity in megafauna would not confer greater protection against human pressure in these areas.

While the human-megafauna link is clear, the mechanisms involved are less so. It has been hypothesized that, because *Homo* originated in Africa and was present in much of southern Eurasia from an early date, local megafauna would have had time to adapt to gradually intensifying hominin predation over a long period of time, whereas in Australia, the Americas, or on islands megafauna would have faced highly developed groups of *H. sapiens* with no prior adaptation, leading to severe extinctions (e.g., Martin, 1967; Sandom et al., 2014a). It is unclear what form such adaptation would take, but such responses are seen in modern "human predator" systems, for example decreased body size and increased reproductive capacity (Darimont et al., 2009). Traits conferring vulnerability to hunting, such as slow movement, might also have been more common outside Africa and Eurasia (Johnson, 2009). Here, such susceptible species might have been driven to extinction much earlier (e.g., Martin, 1966), potentially allowing for diversification in more human-tolerant groups, such as smaller bovids. As mentioned earlier, there are Early-Middle Pleistocene simplification dynamics involving both global extinctions and regional extirpations in megafaunas in Eurasia and Africa. These have been linked to climate change (e.g., Faith et al., 2018; Bibi and Cantalapiedra, 2023; Zhang et al., 2024), but also, alternatively, to pre-*sapiens* hominin impacts beginning as far back as 2–4 million years ago, through the evolutionary insertion of hominins into the large-carnivore guild (e.g., Martin, 1966; Antón et al., 2005; Faurby et al., 2020b; Domínguez-Rodrigo et al., 2021; Dembitzer et al., 2022; Plummer et al., 2023). Among these earlier extinctions, there

is evidence of butchery by hominins, for example *Sivatherium* (Organista et al., 2017), *Deinotherium* (Surovell et al., 2005), and giant geladas (Shipman et al., 1981) in Africa and the large beaver *Trogontherium cuvieri* in Europe (Yang et al., 2019). Similar patterns are also seen in the extinction of large and giant tortoises across the globe. While not strictly megafauna by some definitions, these large reptiles are obviously sensitive to human predation and have a long history of hominin procurement (Rhodin et al., 2015). The largest tortoises disappear first in Africa during the Early Pleistocene, followed quickly by mainland Eurasia, then experience severe extinctions during the late Quaternary in Australia and the Americas, followed by many islands in the Holocene and in recorded history (Rhodin et al., 2015). Additional factors may also have played a role in generating the human-megafauna link, for example higher disease load for *Homo* populations in areas long occupied by hominins due to coevolved parasites and pathogens (Koch and Barnosky, 2006). A potentially underexplored theme is the influence of habitat on modulating human-linked extinction patterns. It has, for example, been suggested that less human-accessible habitats potentially explain some survivals (Johnson, 2002).

Human causation has been questioned based on the hypothesized rapidity of extinctions by relatively small groups of humans in North America (Martin, 1973). While there is clear evidence of rapid, permanent declines in many localities (e.g., Gill et al., 2009; Rule et al., 2012; Adeleye et al., 2023), this so-called "blitzkrieg" version of human overkill is clearly not adequate, with evidence for declines and extinctions occurring over many millenia, particularly at broader geographic scales (e.g., Koch and Barnosky, 2006). Often there is evidence for continent-level overlap between extinct megafauna species and *H. sapiens* across very extended time periods (e.g., Andermann et al., 2020), with megafauna kill sites in South America spanning ~10,000 years, for example (Bampi et al., 2022). Such overlaps have long been recognized in Eurasia, given the late survival and eventual human-driven extinction and extirpation of taxa such as aurochs (*Bos primigenius primigenius*) (e.g., Crees et al., 2016) and northern populations of Asiatic elephant (*Elephas maximus*) and other megafauna in East Asia (Teng et al., 2020). These patterns are further supported by emerging evidence of later survival in relict megafauna populations than formerly thought, for example in northern Eurasia and North America (Murchie et al., 2021; Wang et al., 2021). Long-extended extinction processes are in fact expected under a human impact model given the progressive increase in human population density and socio-technological capabilities (e.g., Ellis, 2015), as well-documented for the mass killing, progressive decline, and eventual near-extinction of ungulates in the Holocene Levant (Bar-Oz et al., 2011). Similar dynamics in population density and culture across the transition from archaic to modern humans also offer an explanation for why only the expansion of the latter caused severe, wide-scale megafauna extinction (aside from the exceptions discussed above), despite the former also being capable megafauna hunters (e.g., Domínguez-Rodrigo et al., 2021; Gaudzinski-Windheuser et al., 2023).

Other points of criticism regarding human causation include a perceived rarity of kill sites and the amount of waste that would be necessary for small human populations to have exterminated megafauna, but both are expected based on taphonomic bias, modern hunter-gatherer analogs, and the relative scarcity of relevant archeological sites (Alroy, 2001, Koch and Barnosky, 2006; Surovell and Waguespack, 2008; Ben-Dor et al., 2011; Wolfe and Broughton, 2020). In summary, while many details of a human-driven model are subject to further investigation, it remains highly consistent with the total evidence (Table 2).

## Ecological consequences

Current megafauna-poor ecosystems are ecologically novel relative to evolutionary baselines since the majority of modern taxa (e.g., plants, invertebrates) originated and evolved with diverse, abundant large-mammal assemblages (e.g., Martin, 1967; Donlan et al., 2006; Fløjgaard et al., 2022; Figure 4). Since large mammals often have disproportionate impacts on ecosystem structure and processes (Bakker et al., 2016; Enquist et al., 2020; Pringle et al., 2023), the loss of megafauna likely has had strong consequences for many ecosystems (Johnson, 2009; Malhi et al., 2022) and has led to major shifts in biotic community assembly (Carotenuto et al., 2016; Lyons et al., 2016; Tóth et al., 2019; Smith et al., 2023). Importantly, the effects of large mammals cannot be fully compensated for by smaller species (e.g.,Lundgren et al., 2024; Trepel et al., 2024) and as such, the changes in impact are expected to have been amplified by the bias in extinctions toward the largest megafauna species (Smith et al., 2018, Smith et al., 2023; Figure 1).

The consequences of losing large-bodied species are intuitive since many ecological processes scale with body size, for example metabolic rate and energy consumption (Enquist et al., 2020), forage selectivity (Bell, 1971; Jarman, 1974; Pansu et al., 2019), movement capacity (Berti and Svenning, 2020), and associated processes like seed (Fricke et al., 2022) and nutrient dispersal (le Roux et al., 2020). However, we have a poorer understanding of other organismal traits, life-history strategies, and behavioral expressions that have been lost and may have had large impacts on community and ecosystem processes (Pringle, 2020). For example, proboscideans, which suffered particularly high extinction rates, have prehensile trunks that, in combination with large body size, have the potential to exert unique impacts on plants and vegetation structure (Haynes, 2012). Another example is seasonal mega-herd migrations of medium to large-bodied herbivores, now only existing in a few places on Earth, where they have overriding impacts on ecosystem dynamics (e.g., Subalusky et al., 2017; Owen-Smith et al., 2020; Anderson et al., 2024). Such migrations were much more common before human-driven extirpations (e.g., Spinage, 1992; Bar-Oz et al., 2011).

In this section, we evaluate how megafauna affected and continue to affect ecosystem and community processes, drawing on direct evidence from both the past and from contemporary ecosystems. In doing so, we discuss the known and likely ecological consequences of the late-Quaternary megafauna extinctions, with the evidence together indicating that strong losses of large-bodied animals constitute a fundamental re-shaping of terrestrial ecosystem structure and functioning worldwide (Figure 5; see also Smith et al., 2023). The processes by which megafauna affect ecosystems can be grouped within three broad categories: (1) trophic processes that enforce a top-down control on lower trophic levels and ecosystem processes, (2) physical engineering of the abiotic and biotic environment and processes, and (3) the transportation of energy and matter, including nutrients, plant seeds, and smaller organisms.

### Trophic processes

There is a wealth of literature on direct local- to landscape-scale effects of extant large herbivores on vegetation, including vegetation biomass and structure (Asner and Levick, 2012; Davies et al., 2018; Lundgren et al., 2024; Trepel et al., 2024) as well as plant community composition (Bakker et al., 2006; Bakker et al., 2016; Staver et al., 2021). Herbivore pressure can shift plant species assemblages in various ways. Species-specific dietary preferences

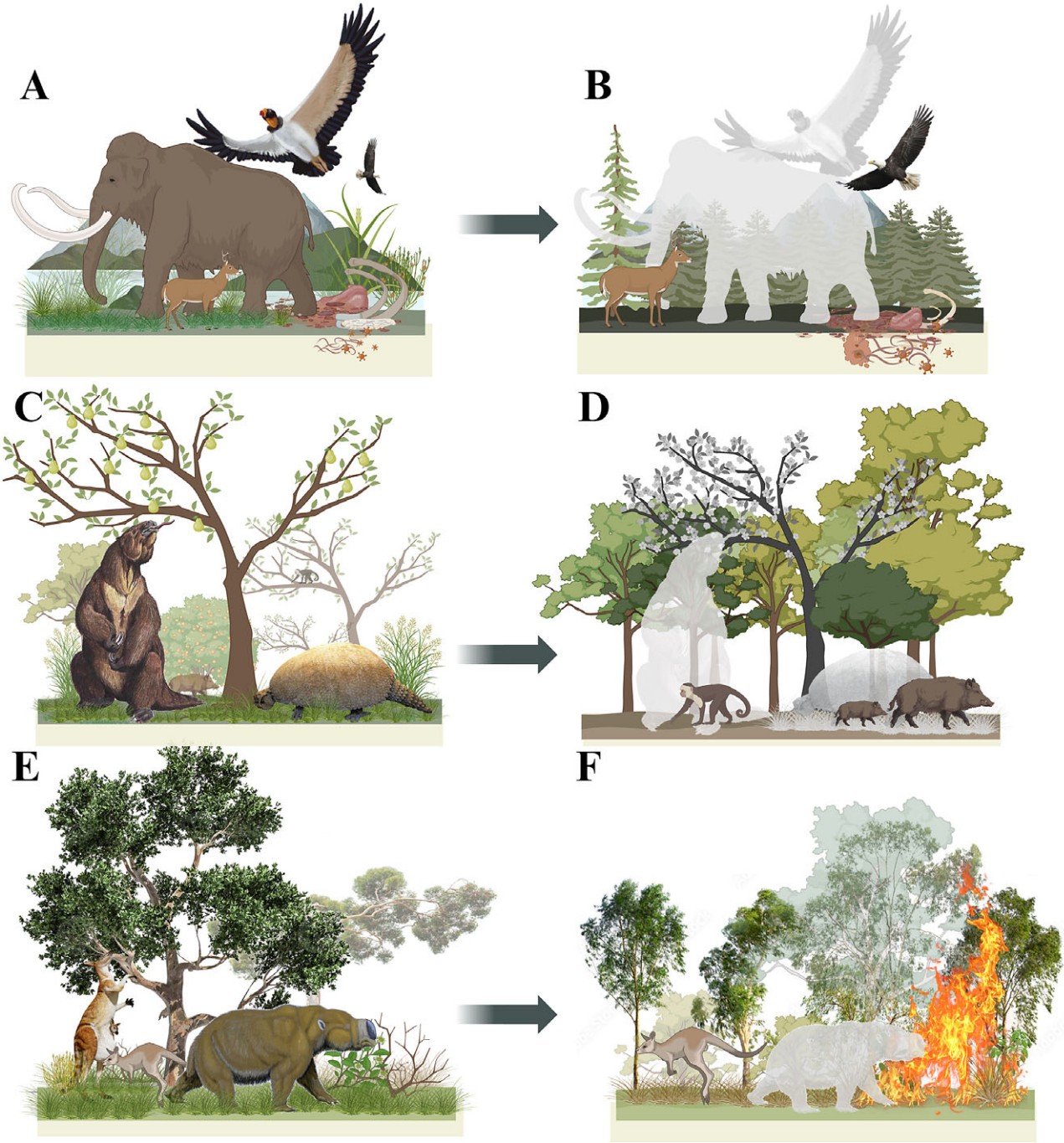

**Figure 5.** Potential impacts of megafauna extinctions on ecosystems. Apart from downsizing faunal biomass, loss of vegetation heterogeneity, and reducing trophic complexity across biomes, megafauna extinctions can also be linked with contemporary ecosystem functioning. Here, three potential cases, all supported by the literature. (A) Across the Americas, big carcasses from the high abundance of proboscideans and other megafauna supported large scavengers (Galetti et al., 2018). (B) Megafauna extinction can be linked to extinctions and extirpations of large scavengers in the region and may have driven changes to the vector-borne pathogen regime (Doughty et al., 2020). (C) In the tropical America, large herbivores such as ground sloths and glyptodonts likely dispersed large-seeded plants long distances and promoted greater vegetation openness through the consumption of plant biomass (Janzen and Martin, 1982; Doughty et al., 2016a). (D) The extinction of megafauna from tropical Americas can be linked to the reduced dispersal of large-seeded plants and structural homogenization of vegetation cover (Doughty et al., 2016a). (E) In Australia, giant kangaroos and diprotodons extensively fed on a wide variety of plants, likely contributing to the maintenance of vegetation diversity. (F) Megafauna extinctions in Australia can be linked to intensified fire regimes in some areas and associated declines in fire-sensitive plant taxa (Rule et al., 2012). (Images: A, HodariNundu; C, D, Bogdanov, WolfmanSF Naturhistorisches Museum Wien [Wikimedia Commons]; E, Dmitry Bogdanov [Wikimedia Commons], Queensland Museum).

will see some species being consumed more frequently than others and shift species composition in favour of the less palatable plant species. However, some plant species are better equipped to deal with herbivore pressure, either by being able to avoid consumption or by being able to tolerate it (Archibald et al., 2019). Herbivores may also shape community composition in more indirect ways, where they can keep the growth of dominant species in check and thereby provide the opportunity for competitively inferior species

to keep a foothold in the community. This disruption of plant competitive interactions links trophic impact to high plant diversity (Koerner et al., 2018), particularly when the feeding disturbance is from large, indiscriminate feeders that are forced to target quantity over quality (Lundgren et al., 2024; Trepel et al., 2024).

Given their current vegetation relations, the late-Quaternary megafauna extinctions are estimated to have had strong effects on vegetation structure and dynamics, not least due to declines in herbivory levels (e.g., Bakker et al., 2016; Doughty et al., 2016a, Pedersen et al., 2023; Davoli et al., 2024; Figure 5). There are palaeoecological studies consistent with strong herbivore impacts on vegetation prior to the late-Quaternary megafauna extinctions, for example high amounts of open and semi-open vegetation alongside high densities of large herbivores in Last Interglacial Britain (Sandom et al., 2014b). Disturbance-linked vegetation types occurred widely across Europe in the same time frame (Pearce et al., 2023), despite the generally warm, productive climate, consistent with high estimated densities of large herbivores (Davoli et al., 2024). There are also palaeoecological studies of other ecosystem types, for example the mammoth steppe, supporting at least some direct role of the late-Quaternary megafauna losses in the collapse of this forb- and grass-rich biome type (e.g., Murchie et al., 2021; Magyari et al., 2022). Other cases of vegetation transformations associated with megafauna extinctions come from eastern North America (Gill et al., 2009) and Australia (Rule et al., 2012), but strong changes do not always seem to have happened (Johnson et al., 2016), and their generality, detailed characteristics, and the mechanisms involved need to become better understood.

Large carnivores can have strong impacts on the population dynamics (e.g., Le Roux et al., 2019) and behaviour (e.g., Le Roux et al., 2018) of specific herbivore species. However, predation in many contexts does not reduce overall herbivore biomass (Mduma et al., 2001; Le Roux et al., 2019) or, by extension, consumption of plant biomass (Beschta et al., 2020; Hobbs et al., 2024). Notably, megaherbivores (body mass ≥1,000 kg) are essentially immune to predation due to their large size and strength, have strong impacts on vegetation, and were omnipresent across all continents except Antarctica until the end-Pleistocene and Early Holocene (Owen-Smith, 1987). Although dietary reconstruction suggests that machairodont cats in some cases likely hunted juvenile megaherbivores (DeSantis et al., 2021; Smith et al., 2022), they also indicate that megaherbivores often formed only a small proportion of the diet (e.g., Coltrain et al., 2004; Bocherens, 2015), and population limitation was unlikely (but see Van Valkenburgh et al., 2016). One contributing factor may have been that various megaherbivores also attained larger body sizes than their present-day relatives, for example proboscideans (Larramendi, 2016). Macroecological analyses support a dominant role of megaherbivores in megafauna community assembly relative to carnivores and mesoherbivores, alongside carnivore effects on the latter (Mondanaro et al., 2017).

Research from Africa shows that even restoring megacarnivores such as lions (*Panthera leo*) does not lead to decreases in overall herbivore biomass, but instead shifts it to the larger species, including the bigger non-megaherbivores (Le Roux et al., 2019). The wolf reintroduction to Yellowstone National Park provides a similar case. Here, the re-establishment of wolves, alongside resurging numbers of other large carnivores, coincided with a decline in the elk population (*Cervus canadensis*), albeit likely enhanced by human hunting (Hobbs et al., 2024). However, this decline is now increasingly compensated by rising numbers of American bison (*Bison bison*), a species rarely predated by wolves, despite high levels of culling (Beschta et al., 2020; Hobbs et al., 2024). Along

similar lines, the largest surviving wild herbivore in Europe, European bison (*B. bonasus*) does not experience significant predation by wolves or other carnivores where they co-occur in eastern Europe (Okarma, 1995). Importantly, while overall herbivory may not be down-regulated by predation in assemblages with diverse megafaunas, such predation-linked shifts toward larger herbivore species are expected to lead to changes in forage species selection and spatial patterns of impact. Smaller species are more likely to exhibit spatial responses to predation risk (e.g., Davies et al., 2021; Lundgren et al., 2022), while larger species are more likely to include low-quality food in their diet (e.g., Daskin et al., 2023). Predation-driven shifts toward larger herbivore species, whose bulk requirements force them to feed more indiscriminately, tend to promote local plant diversity (Lundgren et al., 2024; Trepel et al., 2024), as mentioned earlier.

Beyond herbivory and predation, megafauna also cause other trophic interactions via their generation of various biotic microhabitats and resources, notably their dung, living bodies, and carcasses, with numerous dependent biota (e.g., Galetti et al., 2018). Reflecting this, such dependent groups have seen extinctions as well as declines, for example scavenging birds and dung beetles (e.g., Galetti et al., 2018; Schweiger and Svenning, 2018). In addition to interactions between trophic levels, species also interact within trophic levels in megafauna communities through competition and facilitation, which have cascading impacts on lower trophic levels. A classic example is the successive migration of herbivores in the Serengeti, where smaller animals follow larger animals to benefit from nutritious regrowth (Bell, 1971; Herrik et al., 2023; Anderson et al., 2024). Intra-trophic level interactions among large herbivores have also been inferred from the Cenozoic fossil record (e.g., Mondanaro et al., 2017). Overall, it is clear that the extinction and extirpation of megafauna in the late Quaternary must have directly and severely reduced trophic interactions within mammal communities (Fricke et al., 2022) and with other organism groups (e.g., Galetti et al., 2018), for example through strongly reduced vegetation consumption (Alroy, 2001; Pedersen et al., 2023; Davoli et al., 2024).

Aside from direct trophic impacts on other organisms, megafauna have potentially large indirect effects on ecosystems through the manipulation of natural fire regimes, with large-scale implications for vegetation structure and functioning, potentially even affecting biome distributions and biogeochemical fluxes with Earth-system relevance (e.g., Archibald and Hempson, 2016; Foster et al., 2020; Schmitz et al., 2023; Figure 5). By consuming plant biomass herbivores reduce fuel loads, often reducing local fire frequency and intensity (e.g., Foster et al., 2020). That fire-herbivore interactions can indeed dramatically affect vegetation states is directly evident from the palaeoecological record, which shows several cases across at least three continents where Late Pleistocene losses of megafauna appear to have enhanced fire regimes, in some cases resulting in large vegetation changes (Gill et al., 2009; Rule et al., 2012; Karp et al., 2021). Expectedly, the impact was context-dependent (cf. Foster et al., 2020), and there are regions where the effects on fire regimes were not apparent (Adeleye et al., 2023).

## Physical habitat engineering

Megafauna species play important roles as physical habitat engineers, partly due to their direct and indirect trophic effects (as outlined above), but also via causing non-trophic disturbances and other activities. In the terrestrial realm, there are many

examples of large animals that, by engineering their environment, generate habitats or unlock resources for other species. The digging of water wells, for example, is ubiquitous among elephants and equids (Haynes, 2012; Lundgren et al., 2021). Elephants, specifically due to their large size in combination with a prehensile trunk, have a large potential for engineering environments in additional ways, for example creating hollows by breaking off branches and so generating habitat for hollow-dependent animals (Gordon et al., 2023). Other physical engineering actions that are common among ungulates and other large herbivores, and therefore widespread in areas with large-herbivore communities, include the creation of mud wallows, dust pits, and animal trails, with effects on arthropod communities (e.g., Nickell et al., 2018), fire patterns (Foster et al., 2020), and river ecosystem functioning (Naiman and Rogers, 1997). There is direct paleobiological evidence for such engineering effects by extinct late-Quaternary megafauna, for example proboscidean trackways and large burrows made by ground sloths and, potentially, giant armadillos (Haynes, 2012; Lopes et al., 2017). A likely widespread and understudied form of physical engineering is soil compaction by large herbivores, which, in interaction with soil disturbance, may shape plant communities (Howison et al., 2017; Trepel et al., 2024). Positive feedbacks between repeated grazing and subsequent soil compaction and nutrient input from dung, alternating with ungrazed patches, where bioturbation promotes taller and less palatable plant communities, may drive shifting local mosaics of alternative vegetation states (Howison et al., 2017). The impacts of these and other biogenic modifications can be highly persistent, enhancing the impact of such ecosystem engineering (Albertson et al., 2024). Further, little is known about the potential impacts of megafauna herbivores at natural densities on soil erosion, what the consequences may be for sediment and nutrient loads in rivers, coastal systems and oceans, and how those loadings may affect downstream communities, primary productivity and other ecological processes at local to global scales (e.g., Subalusky and Post, 2019).

### Transportation of organism and materials

Roughly half of the world's plant species are dispersed by animals, mainly birds and mammals, which have the capacity to transport even large seeds (Fricke et al., 2022). Large mammals are also important for smaller-seeded herbaceous plants, including graminoids, with seeds transported endo- and ectozoochorously (e.g., Baltzinger et al., 2019). Plants have evolved obvious adaptations to cling to animal fur or skin, a clear example of how megafauna have acted as a major force on plant evolution. More subtle adaptations may also exist, for example in many herbs (including grasses) foliage serves to attract herbivores that inadvertently ingest and disperse seeds (Janzen, 1984). The most striking example of plant dispersal adaptation to megafauna are the so-called "megafauna fruits," with seeds that are too large to be swallowed or carried by smaller animals or with very large fruits with numerous small seeds (Janzen and Martin, 1982; Guimarães et al., 2008). Where their megafauna dispersers are being lost at present, such species experience reduced regeneration and population declines (Galetti et al., 2018). There are many species with megafauna fruits for which their late-Quaternary partner(s) appear to be extinct, and which may consequently have suffered range contractions or even extinctions (e.g., Janzen and Martin, 1982; Doughty et al., 2016c; Galetti et al., 2018; Onstein et al., 2018), as also directly indicated in the Late Pleistocene ecosystem dynamics of southern Australia (Adeleye

et al., 2023; Figure 5). Fossil dung provides direct evidence that megafauna species such as American mastodon (*Mammut americanum*) dispersed megafauna fruits and many other plant species (e.g., Newsom and Mihlbachler, 2006). Adaptation to dispersal by extinct megafauna is also supported by observations of plants whose regeneration and distribution are promoted primarily by introduced cattle and horses in the Americas (e.g., Janzen and Martin, 1982; Brown and Archer, 1988).

In addition to being the sole bearers of megafauna fruit seeds and transporting large numbers of smaller seeds, the distance by which these seeds can be moved by megafauna is important, not least given the importance of long-distance dispersal events in modulating range and metapopulation dynamics (e.g., Higgins and Richardson, 1999). Potential dispersal distance by terrestrial animals scales with body size, as larger animals have greater stride length, larger home ranges, and longer gut retention times (Berti and Svenning, 2020; Berti et al., 2021; Fricke et al., 2022). Similar impacts may apply to the transportation of small-bodied organisms, including parasites such as ticks (Galetti et al., 2018) but also microorganisms, including disease agents, dung-associated fungi, and potentially mycorrhizal fungi (e.g., Doughty et al., 2020). As a result, movement rates for plants as well as gut-transported microorganisms are estimated to have been strongly reduced by the late-Quaternary megafauna extinctions (Berti and Svenning, 2020; Doughty et al., 2020; Fricke et al., 2022).

The potential for size-dependent transportation rates and distances by megafauna herbivores also extends to abiotic elements including carbon, a primary resource for arthropods and microorganisms, and plant nutrients such as nitrogen, phosphorus, and essential micro-nutrients. These types of effects are estimated to have declined severely with the late-Quaternary megafauna extinctions (Doughty et al., 2016b). Megafauna contribute to shaping nutrient landscapes by increasing nutrient availability and redistributing them spatially. By consuming plant material, herbivorous megafauna liberate nutrients and reduce the amount of nutrients locked up in structural tissue, which can boost ecosystem productivity (McNaughton et al., 1988; Augustine et al., 2003). The ability of larger animals to digest and extract nutrients from large amounts of lower-quality material enables them to access nutrients locked away in more recalcitrant plant tissue and thus release them in forms more readily available for plant uptake (McNaughton et al., 1988). Importantly, megafauna-driven nutrient distribution can create fundamentally different spatiotemporal patterns from those created by abiotic forces (McInturf et al., 2019). For example, gravity propels water to accumulate nutrients in low-lying areas, gradually transporting it to coastlines, whereas mammals can move nutrients in dung against the forces of gravity uphill and back toward continental interiors (Doughty et al., 2016b). As a noteworthy case, hippopotamus dung is responsible for 76% of total silicon flux along East African rivers, thereby strongly affecting downstream primary production and communities (Schoelynck et al., 2019). In addition to depositing dung, the nutrient composition of an animal's body is notably distinct from the ratios in which they exist in the environment. Carcasses can therefore generate distinct local nutrient hotspots, with impacts on ecological processes such as plant productivity (Towne, 2000) and carrion-associated species (van Klink et al., 2020). These smaller-scale patterns and processes have the potential to translate to larger scales and affect, for example, the climate system, but this remains poorly understood (Malhi et al., 2016; Pringle et al., 2023; Smith et al., 2023).

### Large-scale effects

While there exists a wealth of evidence for smaller-scale patterns and processes with potential to translate to larger scales and affect, for example, the climate system, the occurrence of such effects remains incompletely understood, both in relation to the late-Quaternary extinctions and in context of potential, broad-scale megafauna restoration (Malhi et al., 2016; Pringle et al., 2023; Smith et al., 2023). Of particular interest is the degree to which megafauna-mediated processes maintain biodiversity across scales, and especially how these processes are maintained in a rapidly changing world where both climate and other environmental conditions are moving into novel territory, as well as accelerating human-driven redistribution of species. Also important is the magnitude of impact by megafauna on climate through atmospheric carbon, soot and methane (Archibald and Hempson, 2016), above- and below-ground carbon stocks (Wigley et al., 2020; Kristensen et al., 2022; Malhi et al., 2022; Schmitz et al., 2023; Smith et al., 2023), land-surface albedo (Cromsigt et al., 2018; Malhi et al., 2022), the extent and depth of permafrost (Macias-Fauria et al., 2020), and modulation of fire regimes (Malhi et al., 2022). Although these studies offer insights into the potential influence of megafauna on climate, accurately assessing their collective effects on the Earth's system—across past, present, and future timelines—continues to be a complex challenge.

### Megafauna management under global change

The recent, severe losses of megafauna worldwide and the associated ecological consequences have driven attention to the fact that current ecosystems, even in perceived wilderness areas, are in functionally novel, compromised states relative to a deep-time norm, but also to implications for land management in the Anthropocene (e.g., Martin, 1967; Donlan et al., 2006; Svenning, 2020; Svenning et al., 2024). In this section, we discuss three important such issues, namely the integration of megafauna into restoration actions, the management of alien and perceived alien megafauna, and ecological impacts of livestock. All are important not just for the future of Earth's megafauna, but also broadly for biodiversity and ecosystem functioning in the Anthropocene.

### Megafauna-based trophic rewilding as a restoration approach

Trophic rewilding is defined as a restoration approach that aims to reestablish self-sustaining complex, biodiverse ecosystems by restoring top-down trophic processes, with emphasis on the reestablishment of large animals given their high ecological importance, but widespread elimination (Svenning et al., 2016, 2024). Rewilding more broadly focuses on restoring ecological processes alongside reducing human pressures (e.g., Perino et al., 2019). Rewilding is usually conceptualized as an open-ended, forward-looking approach aiming to promote trajectories of high functionality for biodiversity as opposed to the common focus on fixed ecological targets in conventional restoration (e.g., Perino et al., 2019; Svenning et al., 2024). The need to restore self-sustaining and resilient, yet dynamic ecosystems is especially pertinent given ongoing global change, not least climate change and rising human-mediated globalization (Svenning et al., 2024).

As trophic rewilding is a young field the number of empirical studies on self-identified rewilding efforts remains limited, albeit strongly rising (Svenning et al., 2016; Hart et al., 2023). However, there are many empirical studies on the ecosystem effects and biodiversity outcomes of de facto trophic rewilding, especially in Europe, North America, and southern Africa, often under names such as year-round or extensive grazing, or reintroductions, e.g., with many showing positive biodiversity effects (especially for plants) attributable to bison (e.g, Ratajczak et al., 2022) or feral and semi-feral cattle and horses (e.g., Konvička et al., 2021; Dvorský et al., 2022; Bonavent et al., 2023; Köhler et al., 2023). Further, several long-term cases of megafauna recovery that did not necessarily emerge out of rewilding principles show how trophic rewilding can result in large-scale top-down effects on ecosystems, for example vegetation changes in response to the reestablishment of white rhinoceros (*Ceratotherium simum*) (Cromsigt and te Beest, 2014) and African savanna elephants (Gordon et al., 2023). Further, there is evidence that conserving or restoring large-herbivore assemblages helps protect against negative impacts of global change on biodiversity. Examples include limiting warming-induced woody plant dominance to the benefit of low-growing tundra plants, lichens, and fungi (Post et al., 2023), and reducing invasive alien plant abundance to the benefit of native plant diversity (Guyton et al., 2020; Mungi et al., 2023; Svenning et al., 2024). In addition to generating ecological effects and benefitting other species, trophic rewilding can also play a role in the conservation and restoration of wild megafauna, which are among the most endangered functional groups on the planet (Ripple et al., 2016; Atwood et al., 2020).

### Functional effects of non-native megafauna

While the impact of humans on megafauna has generally been a driver of declines and extinctions, a surprising counter-current has occurred over the last couple of centuries through inadvertent megafauna introductions. At least 22 megafauna species (≥100 kg) have established wild populations around the world, representing 29% of the total remaining megafauna species (Lundgren et al., 2018). Fifty percent of these introduced megafauna are threatened or extinct in their native ranges, suggesting novel albeit contentious conservation opportunities (Lundgren et al., 2018). Collectively, these introductions have restored as much as 15–67% of lost late-Quaternary megafauna species richness in each continent (Lundgren et al., 2018). and have restored 39% of lost functional richness, producing functional compositions more similar to those of the Late Pleistocene than assemblages composed only of native species (Lundgren et al., 2020). Indeed, many introduced megafauna are close functional analogs of extinct species (e.g., introduced wild boar [*Sus scrofa*] bear some similarities to extinct peccaries like *Platygonus compressus*) or are conspecific or nearly so with extirpated species (introduced horses – *Equus ferus* – in the Americas) (Lundgren et al., 2020). Other introduced megafauna are phylogenetically distinct from any prehistoric megafauna but still share trait resemblance, for example for Australia there is no species more ecologically similar to the extinct marsupial *Palorchestes azael* than the introduced dromedary *Camelus dromedarius*, based on available functional traits (Lundgren et al., 2020).

Despite their similarity to extinct species, the effects of introduced megafauna are generally considered ecologically harmful and aberrant relative to native megafauna, thereby justifying eradication and control programs, even when introduced megafauna are the same species as Late Pleistocene native species (e.g., wild horses in North America). However, meta-analyses of introduced and native megafauna impacts have found no evidence that nativeness shapes their effects on plants (Lundgren et al., 2024) or on other dimensions of ecosystems, such as small mammals and birds or soil

and plant nutrients (Trepel et al., 2024). This suggests that our understanding of introduced organisms would benefit from critical re-assessment and increased transparency about how we assign "harm" to the effects of organisms and that a dogmatic emphasis on nativeness may be unhelpful to broader conservation goals.

Further research on the impacts of introduced megafauna, especially at broader scales (Trepel et al., 2024) is necessary to understand if and how these organisms restore lost or novel beneficial ecological processes. Some initial research in this domain supports that introduced megafauna may have positive ecological effects. For example, a long-term before-after control-impact study on introduced wild water buffalo in Northern Australia found that buffalo grazing reduced wildfire frequency and severity and increased tree establishment and growth rates (Werner, 2005). Feral pigs, among the most widely introduced and persecuted megafauna species, can increase native plant species richness by controlling competitive dominants (Cushman et al., 2004; Cuevas et al., 2020; Hensel et al., 2022); increase food availability for avifauna (Natusch et al., 2017) and increase nutrient availability and thus tree growth rates (Lacki and Lancia, 1986). Feral equids increase water availability in drylands through well-digging (Lundgren et al., 2021) and maintain open water habitats in desert wetlands, with their removal linked to the extinction of endemic fish populations (Kodric-Brown and Brown, 2007).

Studies on the effects of introduced megafauna should take widespread predator persecution into account, which can modulate the impacts of small- to medium-sized large herbivores, native or non-native (Wallach et al., 2015). In North America, protected cougar populations are major predators of feral horses (Andreasen et al., 2021) and can limit population growth if in habitat with sufficient ambush cover (Turner and Morrison, 2001). Likewise, feral donkeys respond to cougar predation by reducing their activity rates at desert wetlands, with associated reductions in their impacts on wetland vegetation (Lundgren et al., 2022). While conservationists readily recognize that the undesirable impacts of native megafauna, such as heavy browsing by deer (Côté et al., 2004; Martin et al., 2020) are a result of apex predator extirpation or persecution, conservation has been slow to accept that the same could be true for introduced megafauna. However, conversely, it is important to avoid anthropogenic low-density baselines for large herbivores (Fløjgaard et al., 2022). Further, it is important to note that under non-defaunated circumstances there are usually herbivores present which do not experience top-down regulation but are bottom-up regulated by food resources and abiotic stressors (as discussed earlier).

### *Functionality of domestic livestock*

Global declines of wild megafauna have been followed by large increases in domestic livestock densities (Bar-On et al., 2018). Further, in areas such as the Amazon and Chaco regions clearing of natural vegetation to make space for cattle production is rampant to this day. In defaunated systems domestic megafauna may, under certain conditions, replace some of the ecological functions and processes previously provided by wild megafauna. However, there is a need to establish greater clarity on the ecological functionality of cattle (and other livestock) and the ecological contexts under which they may manifest, not least because a global livestock farming lobby claims high-density and production-focused cattle ranching as a means to ecosystem restoration and climate change mitigation (Hawkins et al., 2022). Importantly, there are key differences between wild megafauna and domestic livestock. First, the effects

of domestic livestock are a function of husbandry practices, in addition to essential aspects of the animal's biology. Modern husbandry practices, particularly in affluent countries where the cost of labor is high, generally involve hands-off approaches to grazing including the removal of apex predators, fencing, the maintenance of female-only herds, yearly removals of livestock to feedlots, and inoculation against disease. Notably, when livestock is kept indoors or in small, densely packed pens in industrial agriculture, any natural ecological functionality is lost. Even in more extensive settings, such as with year-round, low-density grazing on unimproved pastures, the situation is complex. High stocking densities (>1 livestock/ha) are associated with generalized negative effects on animal biodiversity (Sartorello et al., 2020). Husbandry practices can lead to stationary herds of livestock, for example, by providing supplementary feed, water or mineral licks at fixed locations. Movement is often further restricted by fencing, which can fragment other wild megafauna populations (Jakes et al., 2018) and may prevent livestock from performing key megafauna functions such as nutrient and seed dispersal. Finally, modern livestock production also involves shipping animals off-site for slaughter and processing and the disposal of carcasses. This may constitute a form of nutrient-mining in grazed landscapes where the amount of minerals removed may parallel primary sources like atmospheric deposition or weathering rates (Abraham et al., 2021). In regions with low nutrient deposition or low weathering rates, this could have deleterious effects on potential primary productivity, even if there is appropriate management for grazing impacts.

Despite the differences between livestock ranching and free-roaming wildlife, livestock in combination with other human impacts may maintain certain ecological processes that were formerly sustained by wild megafauna (Gordon et al., 2021). For example, traditional extensive livestock practices in Europe were able to generate and maintain open vegetation and associated biodiversity in the Holocene (Navarro and Pereira, 2012) and can still do so both there (Troiano et al., 2021) and elsewhere, for example in the Argentinean Pampas (Aranda et al., 2023) and Brazilian Cerrado (Durigan et al., 2022). This can result in reduced dominance of competitive plant species, resulting in greater stability in grassland productivity (Campana et al., 2022) and increased diversity in plants and other taxa such as birds (Frutos et al., 2020). Similar to wild megafauna, livestock can also reduce fire frequency, intensity, and size in grassy ecosystems around the world (Probert et al., 2019; Durigan et al., 2022), and likely also in Mediterranean ecosystems. It has been known for decades that livestock can disperse plant seeds adapted to dispersal by wild megafauna (e.g., Janzen and Martin, 1982; Bruun and Fritzbøger, 2002; Auffret et al., 2012). However, different livestock species can have different effects on, for example, plant species richness, and more importantly, domestic populations may benefit ecosystem heterogeneity and species diversity less than feral populations, as has been shown for horses (Mutillod et al., 2024). While it is clear that single-species livestock systems do not generate the same diversity of ecosystem impacts as a diverse wild megafauna assemblage, these results suggest that extensive livestock practices, under certain conditions, can partly replace lost ecosystem functionality (Gordon et al., 2021).

At the same time it is important to recognize that while traditional herding practices can be compatible with some forms of wildlife conservation, especially with smaller herbivores (e.g., Fynn et al., 2016; Herrik et al., 2023; Simba et al., 2024), larger wildlife species such as buffalo and elephant are likely to experience competition from livestock grazing (Herrik et al., 2023), especially at

high cattle densities (Wells et al., 2022). Finally, driven by socio-economic dynamics, extensive livestock production is itself disappearing in many parts of the world, with a tendency toward commercialization and intensification of the remaining livestock production. For example, in the Mediterranean region of Europe, grazing land has been steadily abandoned over recent decades (e.g., Navarro and Pereira, 2012). To replace livestock-mediated ecosystem services such as reducing wildfire risk, a re-expansion of wild megafauna has been advocated (San Miguel-Ayanz et al., 2010).

Identifying ecosystem restoration approaches for pastoralist and livestock ranching systems is likely to become increasingly important as meat consumption is expected to increase due to rising affluence and human population (Revell, 2015), despite a strong potential for sustainability in shifting toward highly plant-based diets (e.g., Sun et al., 2022). Moreover, societies in semi-arid and arid landscapes that are unsuitable for crop production are likely to maintain dependence on livestock as a source of subsistence and income, which is likely a more ecologically friendly method of food production than irrigation agriculture in these environments. Therefore, an important avenue of future research – complementing research on how to reduce meat consumption globally – would be on how to preserve and rejuvenate modern husbandry practices that facilitate natural megafauna functions while mitigating negative impacts on biodiversity (including wild megafauna) and ecosystems, for example, by mixing extensive livestock grazing with wildlife (e.g., Herrik et al., 2023; Simba et al., 2024), as already suggested by Martin (1967). Further, replacing ruminant livestock with non-ruminants would also substantially reduce greenhouse gas emissions (Cromsigt et al., 2018).

## Conclusion

The severe losses of large-bodied animals in the near prehistory, notably the Late Pleistocene and Early to Middle Holocene, have stimulated interest and scientific debate for decades and continue to do so. However, much evidence on patterns and drivers has emerged in the last 20 years, and our understanding is now much clearer (Table 2). The strong, size-biased megafauna extinctions across the last ~50,000 years are a global pattern, even affecting sub-Saharan Africa albeit less strongly than elsewhere (Figure 1), and affecting all major biomes, from the Arctic to the tropics (Figure 2). Further, it is a unique event for the whole Cenozoic, that is, the last 66 million years (Figure 4), demanding a cause distinctive to the late Quaternary. As it affected different landmasses at different times, occurred across an extended time frame within larger regions, and affected groups that were phylogenetically very disparate, explanations relying on an extraterrestrial impact or a disease are untenable (Koch and Barnosky, 2006; Holliday et al., 2023). The Late Pleistocene saw massive climate changes, and these continue to be put forward as a potential explanation for the megafauna extinctions. However, these changes were no more severe than earlier in the Pleistocene, with the earlier climatic dynamics not causing selective, global megafauna losses. Further, our review shows that there is weak or no support for any major influence of climate on the late-Quaternary extinctions, neither by coarse-grained macroecological patterns nor by detailed spatial/temporal and mechanistic evidence, with much evidence directly against a climatic cause. Conversely, there is strong, cumulative support for direct and indirect pressures from behaviourally modern humans as the key driver. The global expansion of *Homo sapiens* is indeed unique to the late Quaternary, and there is much evidence for the sophisticated megafauna hunting capabilities in even early behaviourally modern humans in various regions. Crucially, the extinctions exhibit strong links in severity to general human biogeography and are spatiotemporally linked to modern humans at the global level and, with increasing evidence also at finer scales, with extinction concentrated at the arrival of *H. sapiens* or happening later, as expected from ongoing socio-technological development (cf. Ellis, 2015). Interestingly, there is emerging evidence that the initial onset of the extinctions may have occurred prior to the Late Pleistocene in regions where pre-*sapiens* hominins would also have exerted pressure on megafauna, albeit to a lesser extent. However, this remains contested, and in the context of the broader Plio-Pleistocene, it is certainly likely that climatic stress and associated declining vegetation productivity played a role in these limited early megafauna losses (e.g., Bibi and Cantalapiedra, 2023), as also seen in other organism groups in various regions (e.g., Raffi et al., 1985; Svenning, 2003; Magri et al., 2017; Mooney et al., 2017).

A broad range of empirical and theoretical evidence shows that the late-Quaternary extinctions have or must have elicited profound changes to the structure and functioning of terrestrial ecosystems worldwide. As such, the late-Quaternary extinctions represent an early, human-driven environmental transformation at a large geographic scale and hence a progenitor of the Anthropocene, where humans have emerged as a major player in planetary functioning. We further conclude that even partial megafauna restoration – notably as implemented via trophic rewilding (Svenning et al., 2016, 2024) – is likely to have positive effects on biodiversity and ecosystem functioning, given that most extant species have evolved in ecosystems with rich large-animal faunas and given the general functional effects of large animals. Still, outcomes are expected to exhibit context- and scale-dependence and be modulated by the functional composition of the fauna, notably its functional trait and trophic composition (e.g., Le Roux et al., 2019; Lundgren et al., 2024). Further, the existing evidence indicates positive potential for biodiversity and ecosystem functioning in wild-living, non-native megafauna in many circumstances as well as, in the context of facilitating biodiversity in production landscapes (land sharing), in non-intensive husbandry practices, designed to facilitate natural megafauna functions. We note that the potential benefits of megafauna restoration remain, regardless of whether the late-Quaternary extinctions were caused by *H. sapiens* or, contrary to the totality of the evidence, past climate stress.

The overall pattern and amount of evidence for *H. sapiens* playing the dominant role in the megafauna extinctions is perhaps one of the clearest, well-supported patterns in ecology (Table 2). However, key open questions concern the detailed mechanisms involved, for example, the role of coextinctions and the role of pre-*sapiens* hominins in limited prior megafauna losses, and much remains to be learnt on the ecological impacts of the extinctions (Figure 5). Further, while there is already much empirical work on trophic rewilding, much more – in differing ecological and societal contexts and via large, well-designed experiments – would be valuable. A key point of attention here should be on trophic rewilding in context of the global emergence of novel ecosystems (Svenning et al., 2024) due to anthropogenic climate change, alien species introductions, and other human environmental transformations (e.g., Guyton et al., 2020; Mungi et al., 2023; Post et al., 2023).

**Open peer review.**  To view the open peer review materials for this article, please visit http://doi.org/10.1017/ext.2024.4.

**Data availability statement.**  No new data are reported in this review.

**Author contribution.** All authors contributed substantially to the conceptualization and writing of the manuscript.

**Financial support.** We received economic support from VILLUM FONDEN via JCS' VILLUM Investigator project "Biodiversity Dynamics in a Changing World" (grant 16549), the Danish National Research Foundation via Center for Ecological Dynamics in a Novel Biosphere (ECONOVO) (grant DNRF173 to J.-C.S.) and Independent Research Fund Denmark | Natural Sciences via the MegaComplexity project (grant 0135-00225B to J.-C.S.).

**Competing interest.** The authors declare no competing interests exist.

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
