## [Editor Report]

Thanks for submitting your paper. Which is a very nice review of a large topic, with rich information content and citations. I am keen that the work should be accepted for publication after some important revisions. Both referees make suggested modifications to the paper. The second referee in particular had more to say.

I am broadly supportive of the comments of the second referee, who has pointed to some potentially misleading uses of language, and some aspects of written tone that could be naive with respect to interpreting fossil data. Importantly, the writing provides lists and counts of extinct species but these are not framed by an expectation under background species extinctions. Also that expectations under a climate model are not always totally clear - hence it is not clear to what extent some observations may be consistent to not consistent with a role of climate in explaining extinctions.

I ask that you look carefully at the referee’s comments, considering how to ensure that the text is circumspect about alternative ideas, and presents alternative viewpoints and interpretations more clearly.

I add a couple of comments here:

-Extinct species lists include a mix of megafaunal taxa > 1 tonne, and lists of species including many that are substantially smaller such as thylacines, native hens and Tasmanian devils. This is confusing because no size info is given for these species. So a reader may expect them to be much larger than in fact they are. Carefully go through the text and ensure that the sizes of various different species are clear.

-The caption to Figure 3 has ‘A’, ‘B’ etc. But the figure itself lacks these.

---

## [Editor Report]

Thanks for resubmitting your manuscript after what was, quite clearly, an attentive and thoughtful process of revision. I have read through your revisions, and response to authors. I am impressed on how conscientious and comprehsnevie you have been, and I am happy for the revised manuscript to be published, pending approval by the senior editors.